# Linkage disequilibrium and population structure in a core collection of *Brassica napus* (L.)

**Mukhlesur Rahman** [1‡]*, **Ahasanul Hoque** [1,2‡], **Jayanta Roy** [1]

**1** Department of Pant Sciences, North Dakota State University, Fargo, North Dakota, United States of America, **2** Department of Genetics and Plant Breeding, Bangladesh Agricultural University, Mymensingh, Bangladesh

‡ MR and AH authors have contributed equally to this work and share first authorship.
* md.m.rahman@ndsu.edu

**Data Availability Statement:** All raw sequence data and variant data are available in NCBI and ENA repository. The accession id for them are PRJNA687906 (https://www.ncbi.nlm.nih.gov/biosample/17159566) and PRJEB42419 (https://

## Abstract

Estimation of genetic diversity in rapeseed is important for sustainable breeding program to provide an option for the development of new breeding lines. The objective of this study was to elucidate the patterns of genetic diversity within and among different structural groups, and measure the extent of linkage disequilibrium (LD) of 383 globally distributed rapeseed germplasm using 8,502 single nucleotide polymorphism (SNP) markers. We divided the germplasm collection into five subpopulations (P1 to P5) according to geographic and growth habit-related patterns. All subpopulations showed moderate genetic diversity (average $H$ = 0.22 and $I$ = 0.34). The pairwise $F_{st}$ comparison revealed a great degree of divergence ($F_{st}$ > 0.24) between most of the combinations. The rutabaga type showed highest divergence with spring and winter types. Higher divergence was also found between winter and spring types. Admixture model based structure analysis, principal component and neighbor-joining tree analysis placed all subpopulations into three distinct clusters. Admixed genotype constituted 29.24% of total genotypes, while remaining 70.76% belongs to identified clusters. Overall, mean linkage disequilibrium was 0.03 and it decayed to its half maximum within < 45 kb distance for whole genome. The LD decay was slower in C genome (< 93 kb); relative to the A genome (< 21 kb) which was confirmed by availability of larger haplotype blocks in C genome than A genome. The findings regarding LD pattern and population structure will help to utilize the collection as an important resource for association mapping efforts to identify genes useful in crop improvement as well as for selection of parents for hybrid breeding.

## Introduction

Rapeseed (*Brassica napus* L., AACC, 2n = 4x = 38), is a recent allopolyploid of polyphyletic origin that evolved from hybridization events between two parental ancestors of *B. oleracea* (Mediterranean cabbage, CC, 2n = 2x = 18) and *B. rapa* (Asian cabbage, AA, 2n = 2x = 20) [1]. Rapeseed genotypes having < 2% erucic acid in seed and < 30 μM glucosinolates in seed meal

www.ebi.ac.uk/eva/?eva-study=PRJEB42419),
respectively.

**Funding:** The study was funded by the U.S.
Department of Agriculture - National Institute of
Food and Agriculture (Hatch Project No.
ND01581).The funders had no role in study design,
data collection and analysis, decision to publish, or
preparation of the manuscript.

**Competing interests:** The authors have declared
that no competing interests exist.

is known as canola, which is the second largest oilseed crops produced in the world after soybean [2]. Canola oil is mostly used in frying and baking, margarine, salad dressings, and many other products. Because of its fatty acid profile and the lowest amount of saturated fat among all other oils, it is commonly consumed all over the world and is considered a very healthy oil [3]. Canola oil is also rich with alpha-linolenic acid (ALA), which is associated to a lower risk of cardiovascular disease [3]. Additionally, canola is utilized as a livestock meal and is the second largest protein meal in the world after soybean [4]. Rapeseed oil has various industrial usages. The rapeseed oil, being simple alkyl esters is the best alternative to diesel fuel. It is more energy-economic and environment friendly than diesel fuel [5]. The high erucic acid content in rapeseed oil also made it suitable for using as lubricants [6] and surfactants [7]. Rapeseed expresses three growth habits, winter, spring, and semi-winter. The spring canola is planted in the early spring and harvested in the late spring of the same growing season [8]. The winter type canola is seeded in the fall, vernalized over the winter to induce flower and harvested in the summer [8]. The semi-winter type is needed for a shorter period of vernalization to induce flower [9].

Rutabaga (*Brassica napus* ssp. *napobrassica* L.) is a cool-weather root crop, grown as table vegetable and fodder for animals [10]. Likewise rapeseed, rutabaga was also derived from natural or spontaneous hybridization between *B. rapa* and *B. oleracea* [11]. European immigrants brought rutabaga to North America [12] from its center of origin Sweden or Finland [10, 13]. Likewise most cruciferous vegetables, rutabaga bears anti-cancer properties [14] and showed considerable variability for morphology, biotic and abiotic stress resistance, seed yield and quality [10, 15].

In the United States of America (USA), the canola production increased 13.5 folds from five years average of 1991–1995 (0.11 m tons) to five years average of 2015–2019 (1.49 m tons) [16]. At the same time, canola oil consumption has increased rapidly in last few years. Statistics shows, though canola production increased, but it is not enough to meet the demand. That's why, every year USA imports huge amount of canola oil (2.50 m tons in 2019) from other countries [2]. In USA, canola production is restricted to north-central region and North Dakota (ND) is the leading canola growing state, where 83% of US canola is grown. The North Dakota State University (NDSU) canola-breeding program could play a vital role in canola economy by developing high yielding varieties, shortening the breeding cycle and expanding canola growing acreage.

NDSU canola breeding has already developed few varieties and handsome amount of breeding populations. However, in recent years, the low genetic diversity of the parental stock is hampering the sustainability of the program. This happened because of same sets of parents has already been crossed in different combinations. The recent origin of *B. napus* as a species and its very recent domestication (400 years ago), as well as selection on few phenotypes (e.g. low erucic and glucosinolate acids, seed yield) also accelerated the low diversity which threatens sustainable improvement of the crop [17]. The narrow genetic diversity might also limit the prospects for hybrid breeding where complementing genepools are needed for the optimal exploitation of heterosis [18]. Therefore, we want to expand the genetic base of NDSU stock by incorporating diversified germplasms to existing collection. To shorten the breeding cycle and maximize genetic gain, we want to use cutting-edge breeding techniques such genome wide association mapping (GWAS) and marker-assisted selection. The knowledge of population structure, genetic relatedness, and patterns of linkage disequilibrium (LD) are also prime requirements for genome-wide association study (GWAS) and genome selection directed breeding strategies [19, 20]. Therefore, it is crucial to study, preserve, and even introduce genetic diversity into rapeseed since the diversity ensures the variability for biotic and abiotic stress resistance, and various agronomical and morphological traits.

We could assess the diversity of a germplasm collection by observing the phenotypic variations or genomic variations among the individuals. Before the advent of marker technology and next generation sequencing technique (NGS), crop diversity was usually assessed based on phenotypic performance. However, phenotyping is time consuming and labor intensive. Moreover, plant growth stages and environmental factors severely affect the phenotyping, results in erroneous prediction [21]. To overcome phenotyping limitations, researchers use DNA-based molecular markers for assessing the genetic diversity. Utilization of molecular markers accelerates the pre-breeding activities, as field phenotyping and pedigree information are not required [22]. Multiple genetic diversity and population structure studies, based on molecular markers [23–27], whole genome resequencing [28], transcriptome and organellar sequencing [29] have already provided information regarding genetic diversity in various *B. napus* collections around the world. However, the genetic diversity of the core collection maintained by the NDSU canola-breeding program has not been revealed yet. That is why; we carried out this research to explore the genetic diversity, population structure level and relatedness among the genotypes and to investigate the linkage disequilibrium (LD) and haplotype block pattern.

## Materials and methods

### Plant materials

A core collection of 383 rapeseed germplasm accessions was used for this study. The core is composed of 67 advanced breeding lines developed by NDSU canola breeding program, 252 germplasm accessions collected from North Central Regional Plant Introduction Station (NCRPIS), Ames, Iowa, USA and 64 varieties collected from different countries. The breeding lines are $F_7$ generation genotypes, obtained by crossing different parents in different combinations. Initially, we collected 500 accessions from NCRPIS and phenotyped them under field conditions. No flowering occurred in case of winter type. Among them, we choose 252 relatively homogeneous genotypes for the core collection. Finally, the core collection was composed of 155 spring, 151 winter, 60 semi-winter, and 17 rutabaga types (S1 Table). The core collection is being and will be maintained through selfing. We grouped the core collection into five subpopulations (P1 to P5) according to their type and origin. Hereafter, we referred the European winter type as subpopulation-1 (P1), Asian semi-winter type as subpopulation-2 (P2), spring type NDSU genotypes (advanced breeding lines) as subpopulation-3 (P3), spring type from different countries other than NDSU breeding lines as subpopulation-4 (P4), and rutabaga type as subpopulation-5 (P5).

### Genotyping and sequencing

DNA was extracted from young leaf tissue, collected from 30 days old plants. We collected three leaf samples per genotype in tubes and flash frozen in liquid nitrogen. Each sample was composed of leaves from three different plant of same genotype. Then we lyophilized leaf tissue and ground it in tubes with stainless beads using a plate shaker. Qiagen DNeasy Kit (Qiagen, CA, USA) was used for DNA extraction (3 samples per genotype) following the manufacturer's protocol. DNA concentration was measured using a NanoDrop 2000/2000c Spectrophotometer (Thermofisher Scientific). The sample that contains good concentration of DNA was kept and other two discarded. Then we prepared the GBS library using *ApekI* enzyme [30]. Finally, Sequencing of the library was done at the University of Texas Southwestern Medical Center, Dallas, Texas, USA using Illumina HiSeq 2500 sequencer.

## SNP calling

SNP calling was done by TASSEL 5 GBSv2 pipeline [31] was used for SNP calling using a 120-base kmer length and minimum kmer count of ten. For alignment of the reads the rapeseed reference genome [32] (available at: ftp.ncbi.nlm.nih.gov/genomes/all/GCF/000/686/985/GCF_000686985.2_Bra_napus_v2.0/) was used. The alignment was done using Bowtie 2 (version 2.3.0) alignment tool [33]. After passing all the required steps, TASSEL 5 GBSv2 pipeline yielded 497,336 unfiltered SNPs. To obtain high quality SNPs, we filtered the raw SNPs using VCFtools [34]. Filtering criteria: minor allele frequency (MAF) $\geq$ 0.05, missing values (max-missing) $\leq$ 50%, depth (minDP) $\geq$ 5, min-alleles = 2 and max-alleles = 2 was maintained to have bi-allelic SNPs. This filtering yielded 53,616 SNPs. To make SNP unlinked, we thinned out SNPs present within 1,000 bp distance. The SNPs that were located outside chromosomes (i.e., position unknown), were removed. As canola is a self-pollinating crop, the SNPs that were heterozygous in more than 25% of total genotypes, were also removed using TASSEL [35]. Finally, we selected 8,502 SNP markers for this study.

## Data analysis

To investigate the population structure, the core collection was differentiated into clusters using STRUCTURE v2.3.4 [36] software. For this purpose, we used admixture model with various combinations of burn-in lengths (5,000 to 100,000) and Monte Carlo Markov Chain (MCMC) lengths (5,000 to 100,000). Each combination was replicated 10 times per K (K1-K10). As we grouped the collection into five subpopulations according to their type and origin, we ran each replication considering genotype assigned to specific subpopulation as well as no subpopulation *i.e.* genotype unassigned to any specific subpopulation. These were done to determine the parameters needed to reach convergence. We used DeltaK approach [37] to determine the ideal number of subpopulations, which was performed by Structure Harvester [38]. We also used median (MedMedK and MaxMedK) or mean (MedMeaK and MaxMeaK) estimators of the "best" K to group the subpopulations into optimum clusters [39, 40]. Ten replicates of Q matrix were assembled using CLUMPP [41] to get individual Q matrix. Structure output was visualized using the Structure Plot v2 software [42]. Principal component analysis (PCA) was conducted by covariance standardized approach in TASSEL [35]. We constructed phylogenetic tree using MEGAX program with 1,000 bootstraps [43] using neighbor-joining (NJ) algorithm. Resulting tree was displayed using FigTree V1.4.4 [44].

We calculated analysis of molecular variance (AMOVA) to partition the genetic variance among subpopulations in Arlequin3.5. To show the divergence, we calculated average pairwise between subpopulations $F_{st}$ values using Arlequin3.5 [45]. Tajima's D value of each group was calculated using MEGAX software [43]. GenAlex v6.5 [46] was used to estimate percentage of polymorphic loci, number of effective alleles, Shannon's information index, expected heterozygosity and unbiased expected heterozygosity of each marker and subpopulation. To visualize SNP density, we developed a distribution plot of SNP using R package CMplot (available at: https://github.com/YinLiLin/R-CMplot). The polymorphism information content (*PIC*) of markers was calculated using software Cervus [47]. To show relatedness among individuals, we calculated kinship (IBS) matrix using software Numericware i [48] on a 1 to 2 scale. The kinship heatmap and histogram were visualized using R package ComplexHeatmap [49]. The correlation between level of relatedness (IBS coefficients) and Shannon's information index (*I*) and diversity (*H*) was calculated in R v3.5.2 [50].

Linkage disequilibrium (LD) pattern of whole collection and different subpopulations were analyzed using PopLDdecay [51]. The mean linked LD was calculated by dividing total $r^2$ value with total number of corresponding loci pair. In this case, $r^2 > 0.2$ was considered only.

Same procedure was followed to calculate mean unlinked LD where $r^2 \leq 0.2$ was considered. Haplotype block analysis was done using PLINK [52] with a window size of 5 Mb. Confidence interval (CI) method [53] was used to identify haplotype blocks with high LD. Haplotype blocks (>19 kb), observed in one subpopulation but not in the other, were considered to be subpopulation-specific block. Haplotype blocks (>19 kb) shared by more than one subpopulation, were considered to be common to corresponding subpopulations.

## Results

### SNP profile

We used 8,502 SNPs, covering 19 chromosomes for this study. The marker density was one per 99.5 kb. Highest number (685 SNPs, 8.06%) markers was situated on chromosome A3 and lowest (236 SNPs, 2.78%) was on chromosome A4. In terms of density, it was highest on chromosome A7 (71.1 kb) and was lowest on chromosome C9 (134.5 kb) (Table 1, Fig 1).

The transition SNPs (4,956 SNPs) was more frequent than transversions (3,546 SNPs) with a ratio of 1.40. The ratio of transitions to transversions SNPs was higher in A genome (1.41) than that of in C genome (1.38). In both genome, G/C transversions were lowest (4.33% and 4.29%), but A/G and C/T transitions occurred in almost similar frequencies (Table 2). The inbreeding coefficient within individuals ($F_{it}$), inbreeding coefficient within subpopulations ($F_{is}$), observed heterozygosity (*Ho*) and fixation index (*F*) of all the markers ranged from -0.45 to 1.00, 0 to 0.73, 0 to 0.57 and 0.40 to 1.00, respectively. The mean Shannon's information index (*I*) of all markers 0.37 with a range from 0.10 to 0.69. The expected heterozygosity (*He*)

**Table 1. Chromosome-wise distribution of SNP markers.**

| Chromosome | No. of SNPs | % SNPs | Start position [a] | End position [b] | Length (Mb) | Density [c] (Kb) |
|---|---|---|---|---|---|---|
| A1 | 440 | 5.18 | 149163 | 35806075 | 35.7 | 81.0 |
| A2 | 392 | 4.61 | 13430 | 34692905 | 34.7 | 88.5 |
| A3 | 685 | 8.06 | 2769 | 49103583 | 49.1 | 71.7 |
| A4 | 236 | 2.78 | 32805 | 23517671 | 23.5 | 99.5 |
| A5 | 413 | 4.86 | 18668 | 31435105 | 31.4 | 76.1 |
| A6 | 448 | 5.27 | 120409 | 36005103 | 35.9 | 80.1 |
| A7 | 384 | 4.52 | 85869 | 27388322 | 27.3 | 71.1 |
| A8 | 281 | 3.31 | 231427 | 27734410 | 27.5 | 97.9 |
| A9 | 541 | 6.36 | 81404 | 45841268 | 45.8 | 84.6 |
| A10 | 305 | 3.59 | 133853 | 22085737 | 22.0 | 72.0 |
| C1 | 445 | 5.23 | 86671 | 50660872 | 50.6 | 113.7 |
| C2 | 589 | 6.93 | 92431 | 68260222 | 68.2 | 115.7 |
| C3 | 651 | 7.66 | 3839 | 80365889 | 80.36 | 123.4 |
| C4 | 634 | 7.46 | 138930 | 70507417 | 70.4 | 111.0 |
| C5 | 366 | 4.30 | 26760 | 44124497 | 44.1 | 120.5 |
| C6 | 414 | 4.87 | 275190 | 45479327 | 45.2 | 109.2 |
| C7 | 518 | 6.09 | 271113 | 62304827 | 62.0 | 119.8 |
| C8 | 383 | 4.50 | 57934 | 46317429 | 46.3 | 120.8 |
| C9 | 377 | 4.43 | 920885 | 51627086 | 50.7 | 134.5 |
| Mean | 447.47 | | | | | 99.5 |

[a] Position of the 1st marker on a particular chromosome corresponding to reference genome

[b] Position of the last marker on a particular chromosome corresponding to reference genome

[c] Density was calculated by dividing the length with the marker number.

## The number of SNPs within 1Mb window size

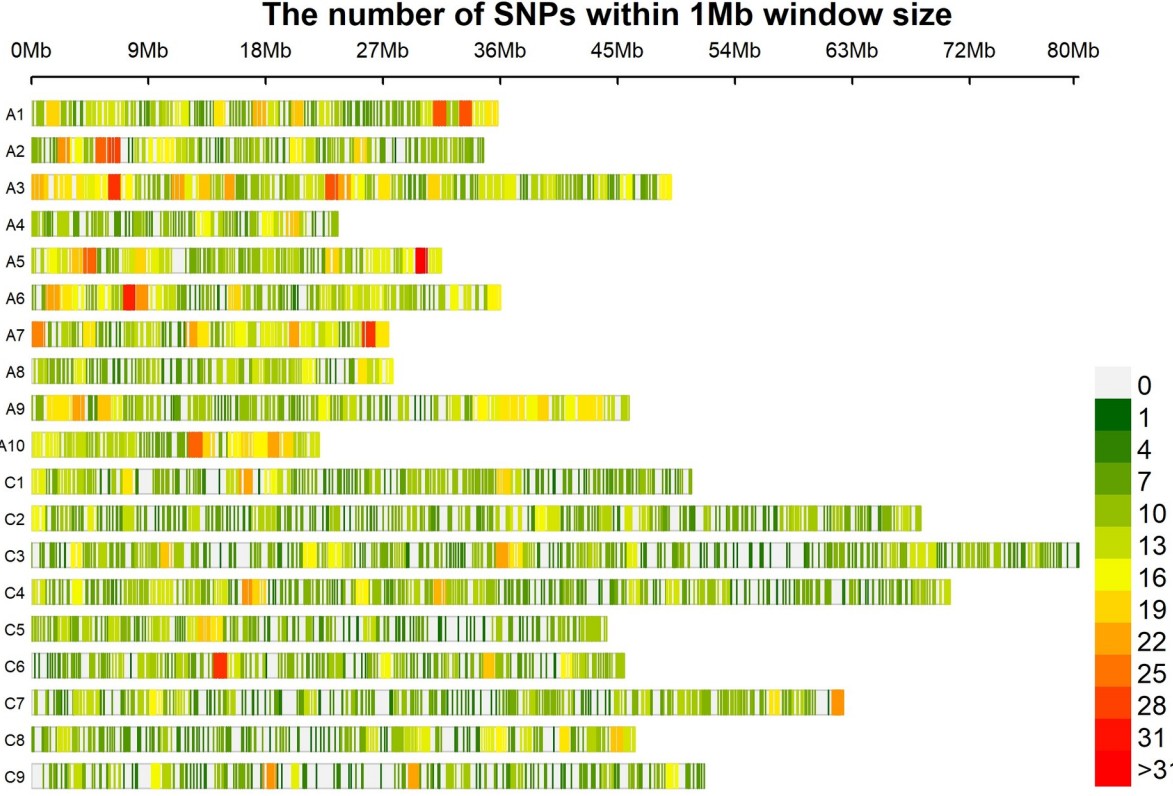

**Fig 1. Chromosome-wise SNP density map.** Frequency of SNPs varies according to color gradient.

was from 0.05 to 0.50 with a mean value of 0.27. The polymorphic information content (PIC) of all markers was less than 0.50 with a mean value of 0.22 (range: 0.05 to 0.37) (S2 Table). Sub-population-wise marker diversity parameters are presented in S3 Table.

## Population structure

We did structure analysis seven times with accessions unassigned and seven times with accession assigned to their type and countries of origin. Delta K approach indicated 3 to 9 clusters

**Table 2. Transition and transversion SNPs across the genome.**

| Genome | SNP type | Model | No. of sites | Frequencies (%) | Total (percentage) |
|---|---|---|---|---|---|
| A | Transitions | A/G | 1195 | 14.06 | 2416 (28.3%) |
| | | C/T | 1221 | 14.36 | |
| | Transversions | A/T | 457 | 5.38 | 1709 (20.1%) |
| | | A/C | 424 | 4.99 | |
| | | G/T | 460 | 5.41 | |
| | | G/C | 368 | 4.33 | |
| C | Transitions | A/G | 1273 | 14.97 | 2540 (29.9%) |
| | | C/T | 1267 | 14.90 | |
| | Transversions | A/T | 496 | 5.83 | 1837(21.6%) |
| | | A/C | 482 | 5.67 | |
| | | G/T | 494 | 5.81 | |
| | | G/C | 365 | 4.29 | |

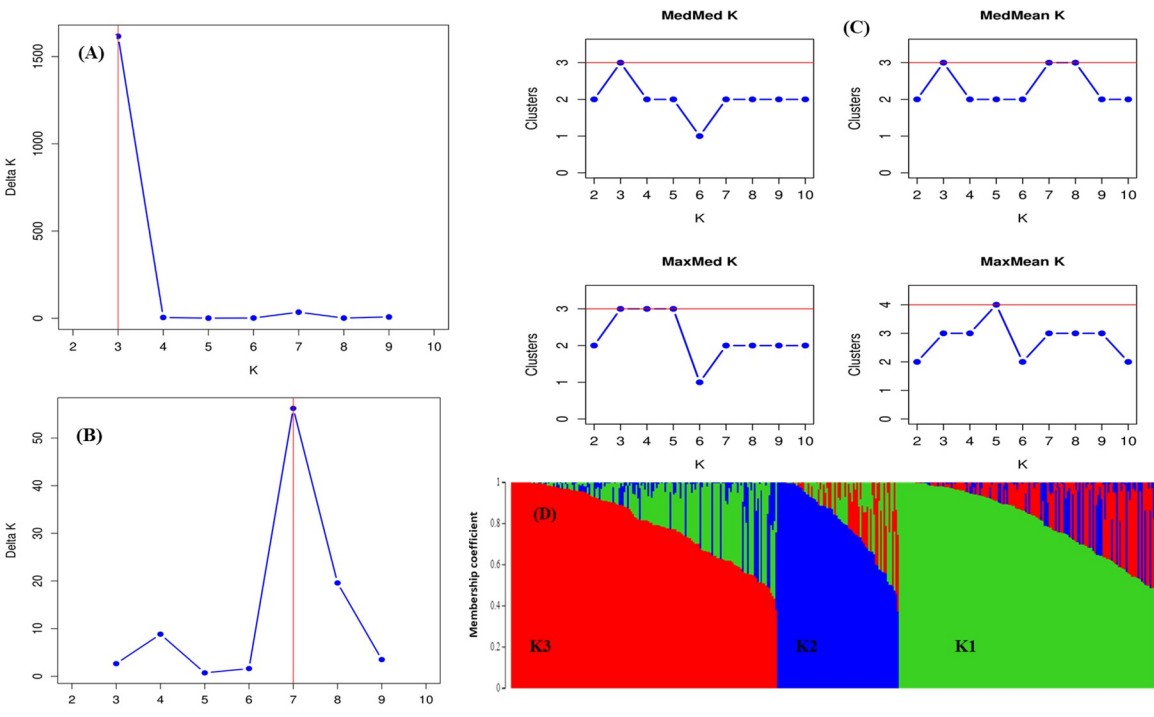

**Fig 2. Bayesian clustering of whole collection using 8,502 SNP markers in STRUCTURE v. 2.3.4.** Graphical representation of optimal number of clusters (K) determined by Evanno's method [37] with genotypes unassigned (A) and assigned (B) to their respective countries, as well as by Puechmaille [39] and Li and Liu [40] method (C). Estimated population structure of 383 rapeseed genotypes on K = 3 (D) using Puechmaille [39] and Li and Liu [40] method.

(Fig 2A and 2B), while four alternative statistics (MedMedK, MedMeaK, MaxMedK, and Max-MeaK) determined following Puechmaille [39] and Li and Liu [40] indicated 3 clusters (Table 3, Fig 2C). For each run, Delta K approach showed differences in cluster number for both conditions: genotypes unassigned or assigned to their respective type and countries of origin. However, opposite scenario was found for MedMedK, MedMedK, MedMedK, and

**Table 3. Clustering of core collection based on Evanno et al. (2005) [37] and Puechmaille et al. (2016) [39] methods using different combinations of burn-in lengths and Markov Chain Monte Carlo (MCMC) lengths.**

| Run # | Burn-in lengths | MCMC lengths | Number of clusters (K) | Number of Reps | Number of populations[α] | | Number of populations[β] | | | |
|---|---|---|---|---|---|---|---|---|---|---|
| | | | | | ΔK (Unassigned) [a] | ΔK (Assigned) [b] | MedMedK | MedMeaK | MaxMedK | MaxMeaK |
| 1 | 5000 | 5000 | 10 | 10 | 3 | 6 | 3 | 3 | 3 | 4 |
| 2 | 10000 | 10000 | 10 | 10 | 8 | 8 | 3 | 3 | 3 | 3 |
| 3 | 20000 | 20000 | 10 | 10 | 8 | 3 | 3 | 3 | 3 | 3 |
| 4 | 20000 | 50000 | 10 | 10 | 8 | 3 | 3 | 3 | 3 | 3 |
| 5 | 50000 | 50000 | 10 | 10 | 9 | 6 | 3 | 3 | 3 | 3 |
| 6 | 50000 | 100000 | 10 | 10 | 9 | 3 | 3 | 3 | 3 | 3 |
| 7 | 100000 | 100000 | 10 | 10 | 3 | 7 | 3 | 3 | 3 | 4 |

[α] The ad hoc ΔK method [31]

[a]Accessions unassigned to any subpopulation

[b]Accessions assigned to subpopulation based on type and origin

[β]The median (MedMedK and MaxMedK) or mean (MedMeaK and MaxMeaK) [33] estimators used to determine the number of cluster (K).

**Table 4. Proportion of admixed and non-admixed accessions per subpopulation based on membership coefficients.**

| Cluster (*K*) | Core collection subpopulation based on type and origin[a] | | | | | Total Number |
|---|---|---|---|---|---|---|
| | **P1: Winter (151)** | **P2: Semi-winter (60)** | **P3: Spring_mixed origin (88)** | **P4: Spring_NDSU (67)** | **P5: Rutabaga (17)** | |
| K1 | 3 | 12 | 58 | 39 | 0 | 112 |
| K2 | 15 | 12 | 8 | 0 | 17 | 52 |
| K3 | 99 | 6 | 1 | 1 | 0 | 107 |
| Admixture [b] | 34 | 30 | 21 | 27 | 0 | 112 |
| In-Cluster | 77.48% | 50% | 76.13% | 59.71% | 100% | 70.76% |
| Admixture | 22.52% | 50% | 23.87% | 40.29% | 0% | 19.24% |

[a] Number of genotypes having $q \geq 0.7$ were assigned to specific cluster.

[b] Genotypes having $q < 0.7$ were considered as admixed genotype.

MaxMeaK statistic i.e., for each run it indicated three clusters. These outputs confirmed that Puechmaille [39] and Li and Liu [40] method was more consistent than Evanno [37] method (Table 3). Structure analysis revealed that 70.76% of genotypes belong to any of the three clusters at similarity coefficient of 0.7 and 29.24% of genotypes are admixed (Table 4, Fig 2D). Spring type accessions fall under cluster-1, whereas winter type European accessions fall under cluster-3. Cluster-2 consists of all rutabaga types and different type rapeseed accessions (Table 4). We performed principal component analysis (PCA) to show the genetic similarity among subpopulations and genotypes. The first two axes explained 21% (PCA1 13.5% and PCA2 7.22%) of the total observed variation (S4 Table). The PCA revealed that rutabaga (P5) and other types having Asian origin make one group, whereas spring type (P3, P4) and European winter type (P1) make two distinct groups (Fig 3). In addition to that, we also constructed unrooted phylogenetic tree based on neighbor joining (NJ) criteria (Fig 4). The output of neighbor-joining (NJ) tree analysis was in line with that of PCA.

## Population diversity

Polymorphic loci percentage was greater than 75% in all subpopulations. P1 bears highest (99%) polymorphic loci, whereas it was lowest in P5 (75%). The diversity (*H*) was lowest in P4 and P5 (0.19) and was highest in P2 (0.25) with an average of 0.22. The Shannon's information index (*I*) ranged from 0.31 (P4 and P5) to 0.40 (P2) with an average of 0.34. The Tajima's D value ranged from -0.70 (P4) to 0.53 (P1) with an average of 0.13 (Table 5).

## Population genetic differentiation

The analysis of molecular variance (AMOVA) revealed that variance among subpopulations covered 24% of total variation and rest of its was covered by among individual variance (Table 6) with a $F_{st}$ and *Nm* value of 0.24 and 1.28, respectively.

We found significant ($p < 0.01$) between subpopulation $F_{st}$ in all combinations. Except combinations P3 and P4 (0.11), P1 and P2 (0.19), we found $F_{st} > 0.20$ for all combinations. The pairwise $F_{st} > 0.30$ was observed between P1 and P5, P3 and P5, P4 and P5 (Table 7).

Kinship analysis showed that the IBS coefficients of the collection ranged from 1.21 to 1.94 with an average coancestry 1.47 between any two canola genotypes (Fig 5, S5 Table). Under P2 subpopulation, almost 50% of total genotypic pairs shows IBS coefficients less than 1.50. In case of other subpopulation, portion of genotypic pairs having IBS coefficient less than 1.50, was very low (Table 8, S1 Fig).

We also performed correlation analysis between mean pairwise relatedness (IBS coefficients) among individuals within subpopulation and Shannon's information index (*I*),

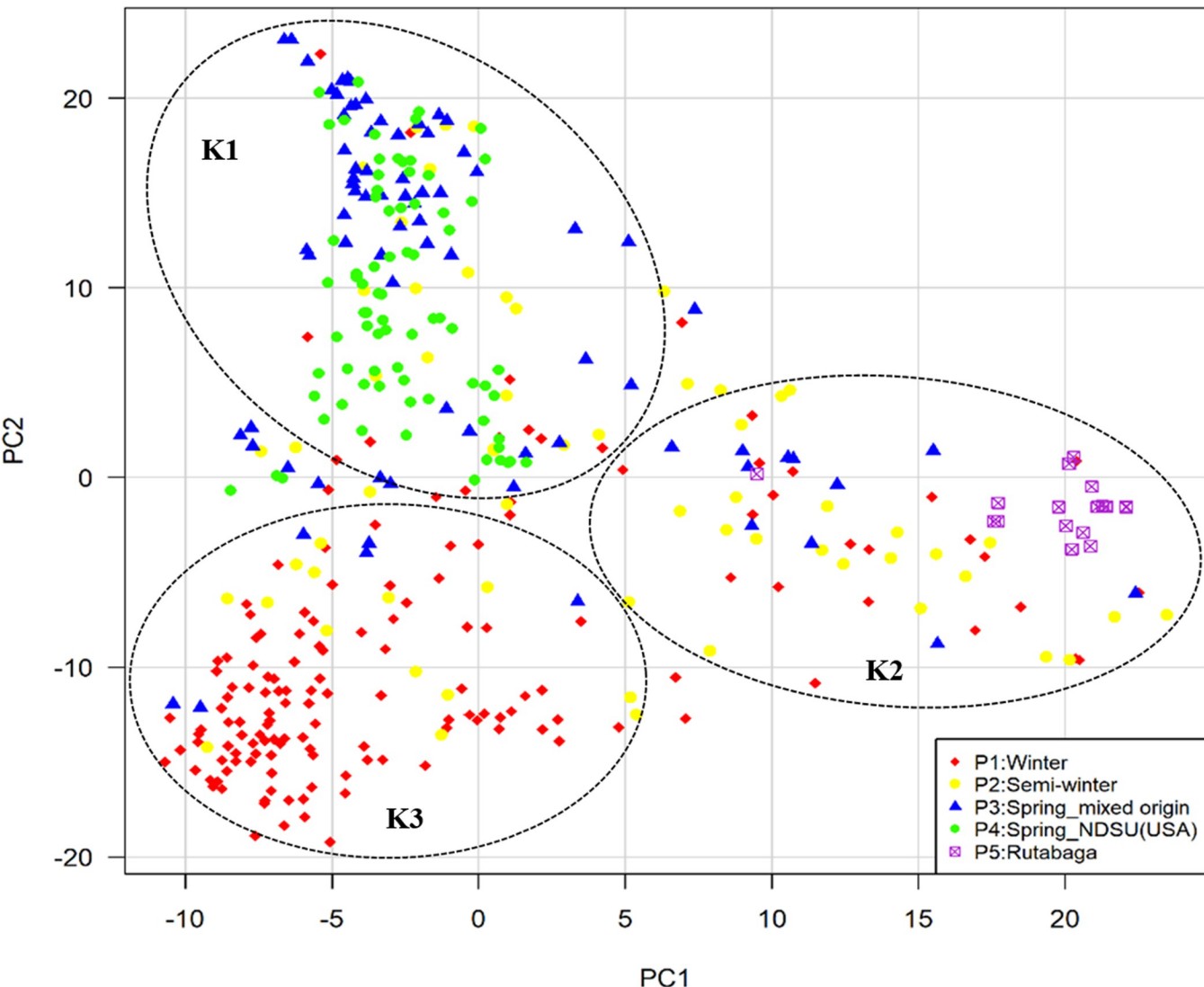

**Fig 3. Principal component analysis of SNP diversity based on genetic distance.** Colors represent subpopulations.

diversity (*H*). The *I* and *H* were significantly and negatively correlated with relatedness (*r* = -0.97, -0.98, and *p* < 0.01), respectively.

## Linkage disequilibrium pattern

Subpopulation, genome, and chromosome-wise linkage disequilibrium (LD) pattern was investigated. LD = $r^2$ values showed inverse relationship with distance i.e., mean LD was high ($r^2$ > 0.22) at short distance bin (0–2 kb) and decreases with bin distance increment (S6 Table). In the entire collection considering both A and C genome, the mean linked LD and mean unlinked LD was 0.44 and 0.02 respectively; and loci pair under linked LD and unlinked LD was 1.81% and 98.20%, respectively. Subpopulation-wise mean linked LD ranged from $r^2$ = 0.41 (P2) to $r^2$ = 0.48 (P1). Subpopulation P5 harbored highest (8.76%) loci pair in linked LD and it was lowest in P1 (1.52%). The mean linked LD, mean LD and loci pair under linked LD was always higher in all cases in case of C genome than that of A genome (Table 9). We also compared the LD decay rate based on distance at which LD decayed to its half maximum

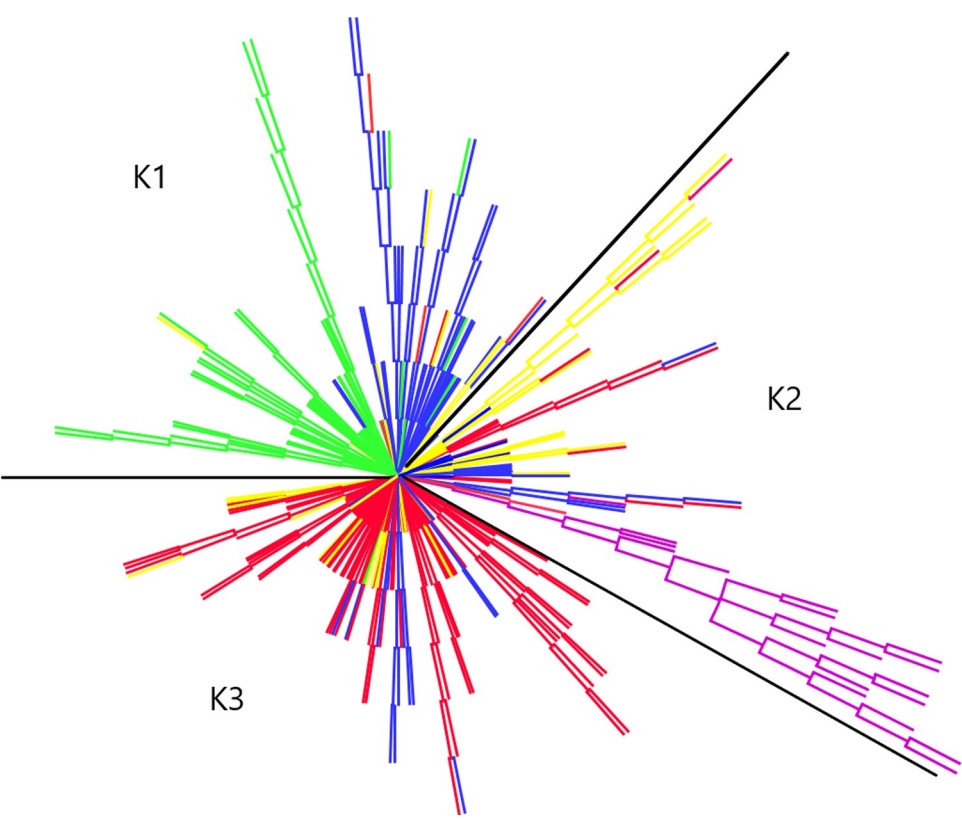

**Fig 4. Phylogenetic tree (unrooted) based on neighbor-joining (NJ) algorithm using information from 8,502 SNP markers based on 1000 bootstraps.** Each branch is color-coded according to genotype belongs to subpopulation P1 to P5. Genotypes were grouped into three clusters by dividing the tree using black solid lines according to structure output.

(half-life), which is the point at which the observed $r^2$ between sites decays to less than half the maximum $r^2$ value. In the whole collection, LD decayed to its half maximum within < 45 kb distance for whole genome, < 21 kb for A genome, and < 93 kb for C genome. In all subpopulations, the distance for LD decay to its half maximum was always higher for C genome than A

**Table 5. Subpopulation-wise diversity parameters.**

| Subpopulations | Polymorphic loci (%) | Na [a] | Ne [b] | I [c] | H [d] | Uh [e] | Tajima's D* |
|---|---|---|---|---|---|---|---|
| P1 | 99.12 | 1.99 | 1.32 | 0.35 | 0.21 | 0.21 | 0.53 |
| P2 | 94.32 | 1.94 | 1.40 | 0.40 | 0.25 | 0.25 | 0.30 |
| P3 | 96.98 | 1.97 | 1.35 | 0.36 | 0.22 | 0.23 | 0.30 |
| P4 | 80.67 | 1.81 | 1.30 | 0.31 | 0.19 | 0.19 | -0.70 |
| P5 | 75.25 | 1.75 | 1.31 | 0.31 | 0.19 | 0.20 | 0.23 |
| Mean | 89.27 | 1.89 | 1.34 | 0.34 | 0.22 | 0.22 | 0.13 |

[a] No. of different alleles

[b] No. of effective alleles

[c] Shannon's information index

[d] Diversity

[e] Unbiased diversity. SE (standard error) was zero in all cases. Indices calculated using 8191 SNPs with GenAlex v. 6.5.

* was calculated with 1000 permutations.

**Table 6. Summary of AMOVA.**

| Sources of variation | d.f. | Sum of squares | Variance components | % of variation | $F_{st}$ | $N_m$ |
|---|---|---|---|---|---|---|
| Among subpopulations | 4 | 130814.6 | 228.5*** | 23.5 | 0.24 | 1.28 |
| Within subpopulations | 761 | 565699.6 | 743.4 | 76.5 | | |
| Total | 765 | 696514.1 | 971.9 | | | |

*** indicates $p < 0.001$ for 1023 permutations.

genome. LD decay rate also varied according to chromosome (S2 and S3 Figs). LD decay was lowest in chromosome C1 (348 kb) and C2 (244 kb), but was highest in chromosome A5 (13 kb) and A1 (16 kb) (S7 Table). LD decayed to its half–maximum within < 29 kb for P1, <45 kb for P2 & P3, <101 kb for P4, and <120 kb for P5. In all subpopulations, LD persisted also longest in all chromosomes of C genome than that of A genome (Fig 6, S7 Table).

We also performed haplotype block (HBs) analysis to investigate LD variation patterns across whole genome. A total 200 blocks covering 18 Mb out of the 976 Mb anchored *B. napus* reference genome [32], were identified. A and C genome contained 67 and 133 haplotype blocks, respectively. The total length of A and C genome specific HBs were 1.8 Mb and 16 Mb, respectively. The total length of HBs varied greatly from chromosome to chromosome. Total HBs length varies from 24 kb on A1 to 901 kb on A9 in A genome and in C genome it varies between 40 kb on C9 to 3,610 kb on C2. The haplotype block (HBs) number and size in C genome chromosome was always higher than that of A genome chromosome (Table 10). We analyzed subpopulation specific and common HBs. We found C genome chromosome bears more subpopulation specific HBs than A genome chromosome (Table 11). We also found some HBs were shared by different subpopulations, but we did not find any HBs blocks that was shared by all five subpopulations (Table 12). The shared HBs were usually located on C genome chromosome. Rutabaga type shared different HBs with other types also.

## Discussion

Genotyping-by-sequencing [30] is one approach to obtain high frequency SNPs. The strategy has been used for population genetic studies, association mapping, and proven to be a powerful tool to dissect multiple genes/QTL in many plant species [54–56]. We obtained 497,336 unfiltered SNPs markers of which 8,502 high quality SNP markers were used for genetic diversity analysis of 383 genotypes. Delourme et al. (2013) [23] conducted genetic diversity analysis in *B. napus* using 7,367 SNP markers of 374 genotypes. However, different marker technologies such as Single Sequence Repeat (SSR), Sequence Related Amplified Polymorphism (SRAP) markers have been used by other researchers for genetic diversity analysis in *B. napus*.

**Table 7. Genetic differentiation among subpopulations.**

| | | | Subpopulation pairwise *Fst* | | |
|---|---|---|---|---|---|
| | **P1** | **P2** | **P3** | **P4** | **P5** |
| P1 | 0 | | | | |
| P2 | 0.19** | 0 | | | |
| P3 | 0.25** | 0.24** | 0 | | |
| P4 | 0.21** | 0.24** | 0.11** | 0 | |
| P5 | 0.34** | 0.24** | 0.34** | 0.39** | 0 |

Diagonal values are pairwise $F_{st}$ comparisons, performing 1000 permutations using Arlequin v. 3.5.

**indicates $p < 0.01$.

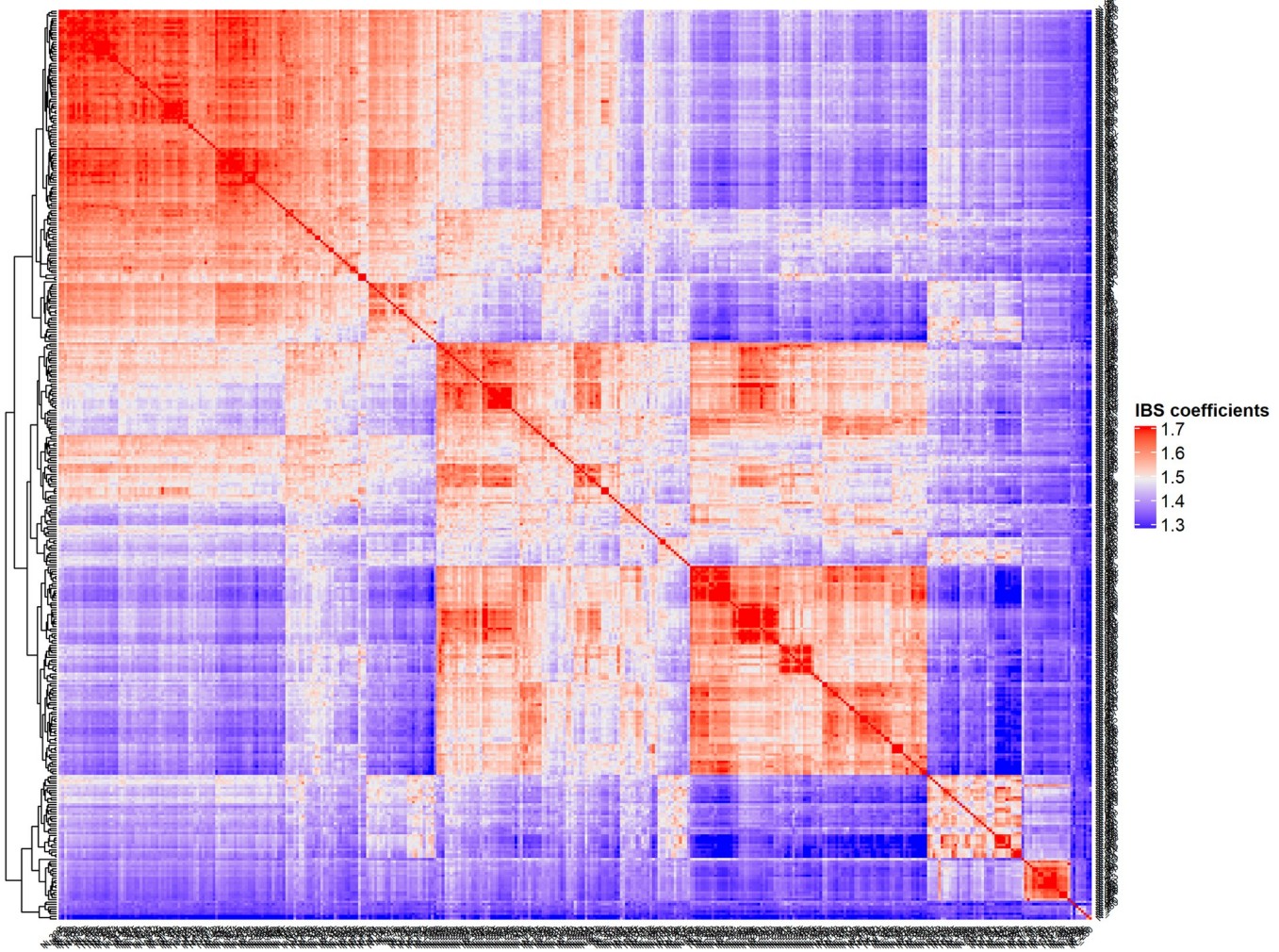

**Fig 5. Heatmap of kinship matrix of entire collection.**

Chen et al. (2020) [57] used 30 SSR markers, Wu et al. (2014) [58] utilized 45 SSR markers, Ahmad et al. (2014) [59] used 20 SRAP markers for genetic diversity and population structure analysis of *B. napus*. Earlier, our group conducted a genetic diversity study of flax using 373 germplasm accessions with 6200 SNP markers [60].

The SNP markers were distributed throughout 19 chromosomes of *B. napus* and the marker density was one per 99.5 kb. This is comparable density to earlier study conducted by Delourme et al. (2013) [23]. Therefore, this marker density provides a sufficient resolution to estimate genome-wide diversity as well as the extent of LD within the genome. This marker density will also help in association mapping studies to identify a causal locus/loci or linked

**Table 8. Summary of subpopulation-wise kinship (IBS) matrix.**

| Subpopulations | Whole collection | P1 | P2 | P3 | P4 | P5 |
|---|---|---|---|---|---|---|
| IBS coefficients range | 1.21–1.94 | 1.40–1.94 | 1.27–1.93 | 1.29–1.93 | 1.46–1.94 | 1.35–1.92 |
| Mean of IBS coefficients | 1.47 | 1.58 | 1.49 | 1.55 | 1.62 | 1.60 |
| Pairs having $\leq$ 1.50 IBS coefficients (%) | 63.9 | 9.6 | 50.7 | 21.7 | 1.1 | 18.0 |
| Pairs having > 1.50 IBS coefficients (%) | 36.1 | 90.4 | 49.3 | 78.3 | 98.9 | 82.0 |

**Table 9. Linkage disequilibrium in the studied collection.**

| Subpopulation | Mean linked LD [a] | Mean unlinked LD [b] | Mean LD [c] | Loci pairs in linked LD (%) | Loci pairs in unlinked LD (%) |
|---|---|---|---|---|---|
| | | | AC_Genome | | |
| Whole collection | 0.44 | 0.02 | 0.03 | 1.81 | 98.2 |
| P1 | 0.48 | 0.01 | 0.02 | 1.52 | 98.5 |
| P2 | 0.41 | 0.02 | 0.03 | 2.65 | 97.4 |
| P3 | 0.45 | 0.02 | 0.03 | 1.94 | 98.1 |
| P4 | 0.45 | 0.02 | 0.04 | 3.98 | 96.0 |
| P5 | 0.43 | 0.03 | 0.07 | 8.76 | 91.2 |
| | | | A_Genome | | |
| Whole collection | 0.33 | 0.02 | 0.02 | 1.34 | 98.7 |
| P1 | 0.38 | 0.01 | 0.02 | 1.12 | 98.9 |
| P2 | 0.32 | 0.02 | 0.03 | 2.02 | 98.0 |
| P3 | 0.36 | 0.02 | 0.02 | 1.41 | 98.6 |
| P4 | 0.40 | 0.02 | 0.03 | 3.45 | 96.6 |
| P5 | 0.38 | 0.04 | 0.06 | 7.06 | 92.9 |
| | | | C_Genome | | |
| Whole collection | 0.50 | 0.02 | 0.03 | 2.21 | 97.8 |
| P1 | 0.52 | 0.01 | 0.02 | 1.83 | 98.2 |
| P2 | 0.46 | 0.02 | 0.04 | 3.27 | 96.7 |
| P3 | 0.50 | 0.02 | 0.03 | 2.35 | 97.7 |
| P4 | 0.48 | 0.02 | 0.04 | 4.41 | 95.6 |
| P5 | 0.46 | 0.03 | 0.08 | 10.57 | 89.4 |

[a] Mean linked LD was calculated by dividing total $r^2$ ($r^2 > 0.2$ was considered) value with total number of corresponding loci pair.

[b] Mean unlinked LD was calculated by dividing total $r^2$ ($r^2 \leq 0.2$ was considered) value with total number of corresponding loci pair.

[c] Mean LD was calculated by dividing total value with total number of corresponding loci pair.

loci that can be further used either in MAS or to pinpoint the causative locus [61] especially for oligogenic traits. However, for polygenic traits such as seed yield, it is better to incorporate more markers for genome wide association studies. The core collection utilized in this study represents mostly adapted lines from various breeding programs. Therefore, sources of variation, markers of interest identified in the collection can be directly used in breeding programs.

We have identified higher frequency of transition SNPs over transversion SNPs that is an agreement with Bus et al. (2012) [62], Clarke et al. (2013) [63], and Huang et al. (2013) [64] in *B. napus*. Higher number of transition SNPs over transversion is also reported in other crop species such as *Hevea brasiliensis* [65], *Camellia sinensis* [66], *Camelina sativa* [67], and *Linum usitatissimum* [60].

To assess the suitability of marker for linkage analysis and diversity, we calculated *PIC* and expected heterozygosity (*He*) of markers [68]. In our research, the *PIC* value is ranged from 0.05 to 0.35 indicating that the markers are modestly informative. The similar lower *PIC* value (0.1 to 0.35) was reported by Delourme et al. (2013) [23] in *B. napus*. The lower *PIC* value is a result of bi-allelic nature of SNP markers and probable low mutation rate [69]. In our study, the *He* value of each marker was always greater than corresponding *PIC* value indicating an average lower allele frequency in the population [68].

## Population diversity and structure

We have identified a moderate diversity (average $H = 0.22$) within the subpopulations. *B. napus* is capable of self-pollination, and little cross-pollination may be occurred by insect.

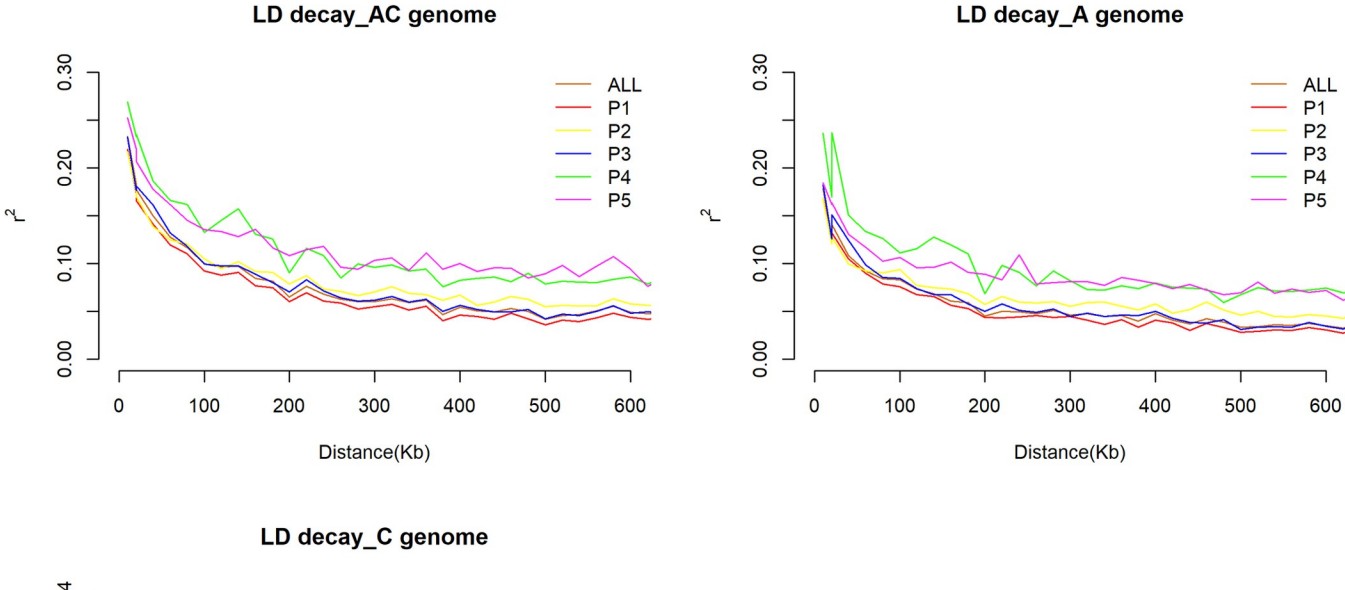

**Fig 6. Linkage disequilibrium (LD) differences and decay pattern among subpopulations.**

Being a mostly self-pollinated crop a low to moderate subpopulation diversity in *B. napus* is expected. Low to moderate diversity was also found in previous studies [70–72]. Along with the reproduction system, one needs to look at evolution and domestication history for explaining low to moderate levels of diversity in *B. napus*. This allopolyploid species originated at Mediterranean coast because of a natural cross between *B. rapa* and *B. oleracea* which occurred approximately 0.12–1.37 million years ago [73, 74]. The domestication of *B. napus* occurred very recently, around 400 years ago with the first rapeseed being most likely a semi-winter type due to the mild climate in the region [75, 76]. Later on, European growers developed the winter and spring type Brassicas through selection for cold hardiness or early flowering to expand its cultivation in further North in the last century [77]. Therefore, the low to moderate diversity in winter and spring *B. napus* can be mostly explained by a recent history of the species, followed by infrequent exchange of genetic material with other Brassicas [23], as well as by the traditional breeding practices selecting for only few phenotypes. In our study, the more diversity in semi-winter type (P2, $H = 0.25$) than winter (P1, $H = 0.21$) and spring (P4, $H = 0.19$) type is supported by its domestication history. The $N_m$ value was greater than one, which indicates that there was enough gene flow among semi-winter, winter, and spring types. These findings also support the evolution of winter and spring types from semi-winter type. In this

**Table 10. Subpopulation-wise number and length of haplotype blocks (HBs) along chromosomes.**

| Chr. | Entire panel | | P1 | | P2 | | P3 | | P4 | | P5 | |
|---|---|---|---|---|---|---|---|---|---|---|---|---|
| | No[a] | Size[b] | No[a] | Size[b] | No[a] | Size[b] | No[a] | Size[b] | No[a] | Size [b] | No[a] | Size[b] |
| A1 | 5 | 24 | 6 | 733 | 2 | 5 | 6 | 557 | 5 | 2080 | 0 | 0 |
| A2 | 8 | 80 | 7 | 96 | 1 | 11 | 8 | 654 | 3 | 15 | 0 | 0 |
| A3 | 6 | 46 | 5 | 29 | 5 | 44 | 8 | 51 | 10 | 927 | 1 | 8 |
| A4 | 5 | 46 | 1 | 27 | 2 | 17 | 0 | 0 | 1 | 6 | 0 | 0 |
| A5 | 7 | 57 | 2 | 13 | 3 | 503 | 5 | 138 | 5 | 427 | 1 | 1 |
| A6 | 9 | 308 | 6 | 564 | 4 | 29 | 5 | 52 | 5 | 51 | 0 | 0 |
| A7 | 8 | 70 | 7 | 72 | 2 | 15 | 7 | 412 | 9 | 1092 | 1 | 13 |
| A8 | 4 | 304 | 1 | 2 | 3 | 21 | 2 | 234 | 6 | 2654 | 0 | 0 |
| A9 | 10 | 901 | 7 | 62 | 14 | 1493 | 6 | 64 | 9 | 1393 | 1 | 21 |
| A10 | 5 | 29 | 6 | 50 | 2 | 21 | 1 | 6 | 5 | 45 | 1 | 1 |
| C1 | 23 | 3099 | 14 | 3314 | 14 | 4638 | 19 | 2947 | 20 | 4295 | 1 | 14 |
| C2 | 26 | 3611 | 19 | 3141 | 9 | 4503 | 22 | 3756 | 16 | 5237 | 7 | 3594 |
| C3 | 14 | 969 | 13 | 930 | 13 | 1480 | 16 | 1603 | 9 | 1070 | 2 | 192 |
| C4 | 16 | 3440 | 15 | 5351 | 15 | 5330 | 14 | 4502 | 13 | 6063 | 4 | 3989 |
| C5 | 9 | 423 | 8 | 410 | 10 | 516 | 7 | 1148 | 7 | 2679 | 1 | 2 |
| C6 | 18 | 1204 | 9 | 972 | 7 | 1191 | 13 | 1206 | 15 | 3357 | 3 | 954 |
| C7 | 17 | 2394 | 17 | 2583 | 7 | 70 | 16 | 2479 | 10 | 1414 | 1 | 13 |
| C8 | 4 | 893 | 5 | 865 | 10 | 1289 | 6 | 941 | 7 | 1143 | 1 | 73 |
| C9 | 6 | 41 | 7 | 49 | 4 | 438 | 7 | 223 | 3 | 11 | 1 | 14 |
| AC Genome | 200 | 17938 | 155 | 19264 | 127 | 21615 | 168 | 20975 | 158 | 33956 | 26 | 8888 |
| A Genome | 67 | 1865 | 48 | 1648 | 38 | 2160 | 48 | 2168 | 58 | 8688 | 5 | 43 |
| C Genome | 133 | 16073 | 107 | 17616 | 89 | 19455 | 120 | 18807 | 100 | 25267 | 21 | 8845 |

[a] Number of haplotype blocks on each chromosome.

[b] Total length of haplotype blocks for each chromosome in kb.

research, Tajima's D value was calculated to identify the extent of availability rare and unique alleles [78]. Recently, the NDSU canola-breeding program developed the P4 advanced breeding lines through crossing different genetic resources including winter, spring, and semi winter types and subsequent selection. This current expansion of P4 was supported by its negative Tajima's D value, which harbors more rare alleles [79]. The subpopulation P1, P2, P3, and P5 showed positive Tajima's D value indicating an excess of intermediate frequency alleles, which may be caused by balancing selection, population bottleneck, or population subdivision. Previously, negative Tajima's D values were found in spring and winter type *B. napus* accessions [80]. The negative correlation between diversity indices (*H* and *I*) and relatedness (average IBS coefficients) indicates that subpopulation differentiation was also due to selfing and genetic drift. Flax [60] and *Arapaima gigas* species [81] also showed same scenario.

To exploit diversity and transgressive segregation, parents from divergent group should be crossed. Pairwise $F_{st}$ statistic, a parameter describing population structure differentiation [82], was estimated among five subpopulations. In the present study all pairwise $F_{st}$ values comprising both low and high values, were statistically significant. Similar results were also found in other studies [80, 83–85]. Lower pairwise $F_{st}$ (0.11) was identified between spring type originated in USA (P4) and spring type originated in other countries (P3). This is reasonably justified as both subpopulations comprise of spring type genotypes and germplasm exchanged occurred between USA and other countries. It also indicating that we will not get higher genetic diversity in population if we use only spring types in the crossing program. But this

**Table 11. Subpopulation specific number and length of haplotype blocks (HBs) along chromosomes.**

| Chr. | P1 specific | | P2 specific | | P3 specific | | P4 specific | | P5 specific | |
|------|-----|-------|-----|-------|-----|-------|-----|-------|-----|-------|
| | No[1] | Size[2] | No[1] | Size[2] | No[1] | Size[2] | No[1] | Size[2] | No[1] | Size[2] |
| A1 | 1 | 683.06 | 0 | 0.00 | 0 | 0.00 | 2 | 1491.24 | 0 | 0.00 |
| A2 | 1 | 39.03 | 0 | 0.00 | 2 | 598.97 | 0 | 0.00 | 0 | 0.00 |
| A3 | 0 | 0.00 | 0 | 0.00 | 0 | 0.00 | 2 | 878.65 | 0 | 0.00 |
| A4 | 1 | 27.02 | 0 | 0.00 | 0 | 0.00 | 0 | 0.00 | 0 | 0.00 |
| A5 | 0 | 0.00 | 2 | 491.27 | 1 | 109.37 | 2 | 408.85 | 0 | 0.00 |
| A6 | 1 | 522.43 | 0 | 0.00 | 1 | 23.04 | 0 | 0.00 | 0 | 0.00 |
| A7 | 1 | 28.42 | 0 | 0.00 | 3 | 392.36 | 5 | 1054.82 | 0 | 0.00 |
| A8 | 0 | 0.00 | 0 | 0.00 | 0 | 0.00 | 3 | 2411.28 | 0 | 0.00 |
| A9 | 0 | 0.00 | 4 | 1409.29 | 1 | 36.87 | 3 | 1339.60 | 1 | 20.50 |
| A10 | 0 | 0.00 | 0 | 0.00 | 0 | 0.00 | 1 | 22.15 | 0 | 0.00 |
| C1 | 4 | 1723.88 | 4 | 3236.73 | 1 | 800.29 | 7 | 2334.07 | 0 | 0.00 |
| C2 | 6 | 1537.92 | 7 | 4464.18 | 6 | 2741.63 | 5 | 3464.31 | 5 | 2796.14 |
| C3 | 2 | 616.65 | 4 | 1024.20 | 4 | 724.95 | 2 | 378.76 | 0 | 0.00 |
| C4 | 4 | 1764.07 | 6 | 2844.27 | 2 | 237.73 | 7 | 5148.84 | 1 | 3439.16 |
| C5 | 1 | 29.99 | 2 | 121.08 | 1 | 771.31 | 2 | 2315.20 | 0 | 0.00 |
| C6 | 3 | 795.86 | 1 | 691.72 | 4 | 782.76 | 4 | 2310.66 | 1 | 715.45 |
| C7 | 3 | 2085.43 | 0 | 0.00 | 4 | 767.17 | 2 | 511.88 | 0 | 0.00 |
| C8 | 0 | 0.00 | 1 | 390.00 | 1 | 885.21 | 4 | 1121.11 | 1 | 73.33 |
| C9 | 1 | 27.36 | 2 | 262.00 | 1 | 32.61 | 0 | 0.00 | 0 | 0.00 |

[1] Number of specific haplotype blocks longer than 19 kb on each chromosome

[2] Total length (kb) of specific haplotype blocks longer than 19 kb on each chromosome.

combination is good for accumulating specific elite trait if the targeted trait is found in members of one and missing from the members of another group. We found spring type (P3 and P4) genotypes are greatly divergent ($F_{st} > 0.20$) from winter and semi-winter type (P1 and P2) genotypes. Utilization of genotypes from these groups in crossing program will broaden the genetic base of developed population results in high heterosis. This potentiality has already been proved as hybrids between the Chinese semi-winter and European (including Canada) spring type exhibited high heterosis for seed yield [86]. The P5 (rutabaga type) showed the higher $F_{st}$ with other subpopulations such as the highest $F_{st}$ was observed between P5 and P4 (NDSU spring type) followed by P3 (other spring type), P1 (winter type) and P2 (semi-winter type). This outcome clearly shows that rutabaga is genetically distinct from spring and winter type canola that is confirmed by previous studies [75, 87, 88]. This distinctness of rutabaga can be exploited through heterosis breeding. Several previous studies have already showed rutabaga as a potential gene pool for the improvement of spring canola [89, 90]. NDSU canola breeding program also utilized winter and rutabaga types in the breeding program for increasing genetic diversity and for improvement of spring canola. AMOVA showed that variation among individual within subpopulation captured greater portion of total variation, than that by among subpopulation. This finding is also supported by earlier researches [56, 72, 91, 92]. This finding supports that within subpopulation genotype from P2, P3, and P1 could be crossed as they showed high diversity ($H > 0.20$) for cultivar development.

Principal component analysis and distance-based population structure analysis such as NJ tree yielded three subgroups in the core collection. Here, we ran structure analysis many times to obtain convergence before the best number of clusters was determined. It was done because previous studies [39, 93] reported that STRUCTURE program did not depict the main clusters

**Table 12. Shared haplotype blocks (HBs) among subpopulation along chromosomes.**

| Chr. | Shared HBs (size and corresponding subpopulation) [a] |
|------|-------------------------------------------------------|
| A1 | 19.991 (P1, P4), 515.231 (P3, P4) |
| A2 | 0 |
| A3 | 0 |
| A4 | 0 |
| A5 | 0 |
| A6 | 0 |
| A7 | 0 |
| A8 | 232.016 (P3, P4) |
| A9 | 19.986 (P1, P2, P4) |
| A10 | 0 |
| C1 | 20.457 (P1,P2, P3, P4), 38.038 (P1, P3), 134.065 (P2, P3, P4), 241.815 (P2, P3), 260.121 (P2, P3, P4), 336.839 (P1, P3), 374.884 (P3, P4), 438.341 (P1, P4), 652.629 (P3, P4), 718.372 (P1, P2) |
| C2 | 20.408 (P1, P2, P4), 28.252 (P1, P4), 164.414 (P3, P4), 729.678 (P1, P3, P4), 781.808 (P1, P4, P5) |
| C3 | 39.715 (P1, P4), 191.098 (P2, P5), 202.569 (P1, P2, P3), 611.558 (P3, P4) |
| C4 | 98.616 (P3, P5), 149.89 (P1, P2, P3), 378.898 (P1, P3), 436.853 (P1, P2, P3), 447 (P1, P3, P5), 601.227 (P2, P3), 867.133 (P1, P3, P4), 1265.92 (P1, P2, P3) |
| C5 | 337.986 (P1, P2, P3, P4) |
| C6 | 96.217 (P4, P5), 136.404 (P3, P4), 142.519 (P2, P5), 237.159 (P3, P4), 308.114 (P2, P4) |
| C7 | 390.085 (P1, P3), 828.502 (P3, P4) |
| C8 | 255.517 (P1, P2), 592.433 (P1, P2) |
| C9 | 171.465 (P2, P3) |

[a] Length (kb) of common NBs longer than 19 kb on each chromosome with their corresponding subpopulation shown in parenthesis.

within a collection. Based on Evanno's ΔK method [37] and MedMedK, MedMeaK, MaxMedK and MaxMeaK statistics [39], structure analysis divided the core collection into three distinct clusters. Cluster-1 contains spring type NDSU advanced breeding lines (P4) and spring type (P3) other than those. This finding is supported by low genetic differentiation ($F_{st} = 0.11$) between P3 and P4 due to sharing of parents by advanced breeding lines from P3. Cluster-3 is solely dominated by European winter type (P1) genotypes. This is also supported by high $F_{st}$ between winter and other types which may be due to geographic barriers between Europe and America, Asia. Cluster-2 contained all rutabaga types as well as other type Asian genotypes which indicates that all types share considerable amount of SNP markers attributing to this cluster. These findings also indicate that there is gene flow among different types, which is also supported by $N_m > 1$. Structure analysis revealed that all clusters contained both non-admixed as well as admixed (share alleles attributed to different subpopulation) genotypes. For broadening genetic diversity of population, non-admixed genotypes should be crossed. However, for improving or introgression of specific traits, admixed genotypes could also be crossed which will reduce the population size required for phenotypic screening. However, structure analysis may overestimate the differentiation among individuals, as the individuals may not share alleles from same ancestors [94]. Since a breeder would like to combine historically never combined favorable alleles, IBS values directs which individuals should be crossed. Low IBS is the best. However, self-pollinated crops exhibit higher kinship values than cross-pollinated crops, as homozygosity increases probability of being identical by state [95]. We found approximately 64% of pairwise coancestry ranged from 1.21 to 1.50. Crossing among genotypes from subpopulation P2 will demonstrate more diversity, than that of other

subpopulations, as most genotypic combinations of P2 shows low IBS coefficients than others do. This finding is in line with the evolutionary history of *B. napus* where semi-winter type is the base population containing more divergence. Gradually this diversity is narrowed down in P3 (spring type, mixed origin) and P1 (winter type), because genotypic pairs belong to P3 and P1 having high IBS values evolved from semi-winter type [77]. Subpopulation P4 exhibited highest number of pairs having IBS > 1.5, which is obvious as these genotypes are advanced breeding lines developed from crossing of same set of parents in different combinations. Genotypic pairs of P5 (rutabaga type) also showed high coancestry may be due to the duplicates which is supported by low genetic differentiation of Nordic rutabaga accessions [27]. We could discard the duplicates during the crossing program.

## Linkage disequilibrium

Linkage disequilibrium can be defined as the correlation among polymorphisms in a given population [96]. The strength of association mapping relies on the degree of LD between the genotyped marker and the functional variant. Linkage disequilibrium analysis provides insight into the history of both natural and artificial selection (breeding) and can give valuable guidance to breeders seeking to diversify crop gene pools [17]. SNPs in strong LD are organized into haplotype blocks, which can extend even up to few Mb based on the species and the population used. Genetic variation across the genome is defined by these haplotype blocks. Haplotypes, which are subpopulation-specific, are defined by various demographic parameters like population structure, domestication, and selection in combination with mutation and recombination events. Conserved haplotype structure can then be used for the identification and characterization of functionally important genomic regions during evolution and/or selection [97]. In addition, the extent of LD needs to be quantified across the genome at high resolution (down to approximately one Kbp) [98]. The information is important for choosing crossing schemes, association studies and germplasm preservation strategies [99–102].

We used markers from across the genome to quantify the LD for the core collection. Low level of LD was evident for each individual subpopulation in A, C, and whole gnome. The low level of LD can be due to multiple factors. First, canola is a partially outcrossing species with an average of 21–30% of cross-pollination [103–105]. The outcrossing occurring in canola leads to more recombination and to a breakdown of haplotype blocks. Secondly, the ancestral history of canola is limited in comparison with other crops, such as rice, common bean, wheat, and corn, restricting the selection of desirable haplotypes during the evolution. In other words, there was no adaptation or domestication pressure on the species, which would lead towards positive selection. Third, the only selection pressure imposed on the species for a relatively short time was breeding. However, the breeding practices were biased towards selection of only few phenotypes. Additionally, the short period under selection pressure might have not been sufficient to select favorable haplotypes in the genome. Fourth, since canola cultivars with different growth habits are compatible there has been always gene flow present between them contributing to the low level of LD. The $N_m$ >1 was observed in this study, which supports this gene flow. Fifth, the restriction enzyme used to develop the libraries for sequencing of the core collection helped in identification of SNPs largely residing in genic regions, which are prone to high recombination, contributing to the low level of LD. Finally, the low level of LD may be due to thinning of markers, as we did not use all markers (53,616) for LD analysis rather used 8,502 markers after thinning. That can be confirmed by analysis the LD using whole marker set in further analysis.

In this study, we have identified that the LD decay in *B. napus* varied across chromosomes of both A and C genomes. In addition, LD in C genome decayed much slower than A genome.

C genome also contained larger haplotype blocks than A genome. This LD patterns are consistent with previous findings [17, 26, 106–108]. The slower LD decay and presence of long haplotype blocks in C genome indicates that high level of gene conservation could have resulted from limited natural recombination or could be exchanged of large chromosomal segment during evolution. In the whole genome, presence of subpopulation specific haplotype blocks suggests that these regions had been experienced selection pressure for specific geographic regions adaptation. In all subpopulations, presence of shorter haplotype blocks in A genome than C genome reveals that *B. rapa* progenitor of *B. napus* containing A genome, which has been used as oilseed crop and probably being used in hybridization process. Sharing haplotype blocks by different subpopulations especially in C genome also confirms its conserved nature. The low level of LD or haplotype blocks has implications for association mapping and a proper experimentation design is necessary for utilizing a reduced set of markers by tagging major haplotypes [109]. Though low LD of A genome requires more markers to pinpoint the location of various QTL, but once a marker is found to be significantly associated with a phenotype, there might be a higher probability of identifying the casual gene than that of C genome.

## Conclusions

This study provides a new insight to select the best parents in crossing plan to maximize genetic gain in the population. The population structure analysis showed a clear geographic and growth habit related clustering. The rutabaga type showed the highest genetic divergence with spring and winter types accessions. Therefore, the breeding strategies to increase the genetic diversity may include generating population from rutabaga and spring crosses, or using rutabaga and winter crosses. The linkage disequilibrium analysis revealed the decay pattern and haplotype blocks in A and C genome. This output will help the breeder to formulate breeding strategies to develop improved cultivars using modern breeding tools by utilizing this collection and SNP markers.

## Supporting information

**S1 Table. List of the genotypes analyzed in this study.**
(XLSX)

**S2 Table. Marker diversity parameters.**
(XLSX)

**S3 Table. Subpopulation-wise marker diversity parameters.**
(XLSX)

**S4 Table (a) Percentage of variation explained by the first 3 axes, (b) Eigen values by axis and sample eigen vectors.**
(XLSX)

**S5 Table. Kinship matrix.**
(XLSX)

**S6 Table. Mean LD values according to distance.**
(XLSX)

**S7 Table. Subpopulation-wise and chromosome-wise LD decay rate (Kb) within each subpopulation.**
(XLSX)

**S1 Fig. Histogram of IBS coefficients.**
(TIFF)

**S2 Fig. Chromosome-wise LD decay rate (Kb) in A genome considering whole collection.**
(TIFF)

**S3 Fig. Chromosome-wise LD decay rate (Kb) in C genome considering whole collection.**
(TIFF)

## Author Contributions

**Conceptualization:** Mukhlesur Rahman.

**Data curation:** Ahasanul Hoque, Jayanta Roy.

**Formal analysis:** Mukhlesur Rahman, Ahasanul Hoque.

**Funding acquisition:** Mukhlesur Rahman.

**Methodology:** Mukhlesur Rahman, Ahasanul Hoque.

**Resources:** Mukhlesur Rahman.

**Software:** Ahasanul Hoque.

**Supervision:** Mukhlesur Rahman.

**Validation:** Ahasanul Hoque.

**Visualization:** Mukhlesur Rahman, Ahasanul Hoque.

**Writing – original draft:** Mukhlesur Rahman, Ahasanul Hoque, Jayanta Roy.

**Writing – review & editing:** Mukhlesur Rahman, Ahasanul Hoque, Jayanta Roy.

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
