## [Decision Letter · Decision Letter 0]

3 Jun 2021

PONE-D-21-09997

Linkage disequilibrium and population structure in a core collection of Brassica napus (L.)

PLOS ONE

Dear Dr. Rahman,

Thank you for submitting your manuscript to PLOS ONE. After careful consideration, we feel that it has merit but does not fully meet PLOS ONE’s publication criteria as it currently stands. Therefore, we invite you to submit a revised version of the manuscript that addresses the points raised during the review process.

We look forward to receiving your revised manuscript.

Kind regards,

Tzen-Yuh Chiang

Academic Editor

PLOS ONE

Journal Requirements:

2a) If there are ethical or legal restrictions on sharing a de-identified data set, please explain them in detail (e.g., data contain potentially sensitive information, data are owned by a third-party organization, etc.) and who has imposed them (e.g., an ethics committee). Please also provide contact information for a data access committee, ethics committee, or other institutional body to which data requests may be sent.

2b) If there are no restrictions, please upload the minimal anonymized data set necessary to replicate your study findings as either Supporting Information files or to a stable, public repository and provide us with the relevant URLs, DOIs, or accession numbers. For a list of acceptable repositories, please see http://journals.plos.org/plosone/s/data-availability#loc-recommended-repositories.

4. Thank you for submitting the above manuscript to PLOS ONE. During our internal evaluation of the manuscript, we found significant text overlap between your submission and the following previously published works, some of which you are an author.

https://bmcgenomics.biomedcentral.com/articles/10.1186/s12864-020-06922-2

Please revise the manuscript to rephrase the duplicated text, cite your sources, and provide details as to how the current manuscript advances on previous work. Please note that further consideration is dependent on the submission of a manuscript that addresses these concerns about the overlap in text with published work.

Reviewers' comments:

Reviewer's Responses to Questions

**Comments to the Author**

1. Is the manuscript technically sound, and do the data support the conclusions?

Reviewer #1: No

Reviewer #2: Partly

Reviewer #3: Yes

2. Has the statistical analysis been performed appropriately and rigorously? 

Reviewer #1: No

Reviewer #2: Yes

Reviewer #3: Yes

3. Have the authors made all data underlying the findings in their manuscript fully available?

Reviewer #1: Yes

Reviewer #2: No

Reviewer #3: No

4. Is the manuscript presented in an intelligible fashion and written in standard English?

Reviewer #1: No

Reviewer #2: No

Reviewer #3: Yes

5. Review Comments to the Author

Reviewer #1: The authors proposed to analyse the genetic diversity present in a collection of 383 accessions of Brassica napus representing genotypes present worldwilde and aims at characterizing the genepool present in US breeding programs. This seems quite interesting and the size of of the populations should be sufficient to draw interesting conclusions. However hypotheses and research questions are not clearly expressed at the end of the introduction, and too many extra data are provided and not used in the mansucript (for instance flowering time data). Moreover, many analyses are performed , but may be too much without really refering to a global strategy and the global aim of each batch of analyses.

Detailled analysis and questions are reported in the attached file

A rapid check at table S1 showed some problems considering geogagraphic origins of different accessions

for instance Lembkes, Krapphauser, fertodi => Germany instead of South Korea

Dramor (Poland) should be Darmor (France) I think

Bienvenu (USA) should be Bienvenu (France)

Jet-neuf (Canada) should be Jet-neuf (France)

.... I didn't check carefully all accessions but these mistakes may lead to inappropriate interpertations of subpopulations composition

Reviewer #2: The manuscript describes an attempt to study genetic diversity and population differentiation inherent in a germplasm assemblage of Brassica napus. Study is of interest for plant breeders engaged in the improvement of this important oilseed crop. Abstract and introduction are very poorly written. The method and material section must include some information about the procedure followed to maintain germplasm lines used for genotyping. In the absence of any information, one would suspect these to be the products of open pollination. Inherent heterozygosity can be a problem for genetic studies in any often cross-pollinated crop like B.napus. Filtering out of a large proportion of heterozygous SNPs may be the cause for the retention of only 8,502 SNPs out 497336 discovered initially. Structure analysis shows lot of admixing. Heterozygosity inherent in the germ plasm will affect all genetic inferences e.g. polymorphic loci, population variation, LD, genetic differentiation and even genetic diversity. It is always better to use homozygous lines. One may not like to use the term core collection for any germplasm assemblage. Authors may like to provide reference or provide details regarding method used to extract the core ( if it is actually a core collection) from a large germplasm base. In spite of these limitations, it was interesting to find clear geographic and growth habit related clustering from STRUCTURE analysis. The manuscript also suffers from narrative coherence.

Reviewer #3: Manuscript PONE-D-21-09997 describes genetic diversity, population structure and LD in Brassica napus - the second major oilseed crop grown worldwide. Here are some comments:

This study does not provide details on sampling (how many plants were sampled for DNA isolation, how SNPs were scored codominant/dominant; how heterozygosity was handled etc). I could not see genotypic data of 383 accessions. Have authors submitted genotypic data to any public database repository or provided in a supplementary Table?

L172-173: Table 1 and Fig 1 does not support this claim.

L191-192: It is worthwhile to provide statistics of transitions and transversions across different chromosomes

L340: GBS is one of the approaches (Illumina SNP, resequencing, sequencing and DArTseq)

L360: There are no wild accessions of B. napus.

L460: Favourable alleles?

Fig 3: How much genetic variation is explained by PC1 and PC2? What these lines indicate?

Fig4: I could not see bootstrap values here. Please label all accessions (name or serial number 1 to 383 and support with supplementary Table)

Figure 5 is not legible. You may move to supplementary information.

Minor comments

L52: Use either rapeseed or canola (it is already described in L49)

L72: There are several studies to support collinearity between Arabidopsis and Brassica (see publications of Parkin and Chris Pires groups, https://doi.org/10.1111/pce.12644 etc)

L527: There are several reports. You meant NSDU accessions?

6. PLOS authors have the option to publish the peer review history of their article (what does this mean?). If published, this will include your full peer review and any attached files.

Reviewer #1: No

Reviewer #2: No

Reviewer #3: No

---

## [Author Response · Author response to Decision Letter 0]

31 Aug 2021

Journal Requirements: 

Response: We have checked and tried our best to format the manuscript according to the PLOS ONE’s style.

2a) If there are ethical or legal restrictions on sharing a de-identified data set, please explain them in detail (e.g., data contain potentially sensitive information, data are owned by a third-party organization, etc.) and who has imposed them (e.g., an ethics committee). Please also provide contact information for a data access committee, ethics committee, or other institutional body to which data requests may be sent.

Response: No restriction

2b) If there are no restrictions, please upload the minimal anonymized data set necessary to replicate your study findings as either Supporting Information files or to a stable, public repo-sitory and provide us with the relevant URLs, DOIs, or accession numbers. For a list of acceptable repositories, please see http://journals.plos.org/plosone/s/data-availability#loc-recommended-repositories.

Response: Datasets are available online.

Response: The study was funded by the U.S. Department of Agriculture - National Institute of Food and Agriculture (Hatch Project No. ND01581).

Response: The funders had no role in study design, data collection and analysis, decision to publish, or preparation of the manuscript.

Response: No salary received.

 Response: Not applicable.

4. Thank you for submitting the above manuscript to PLOS ONE. During our internal evaluation of the manuscript, we found significant text overlap between your submission and the following previously published works, some of which you are an author.

https://bmcgenomics.biomedcentral.com/articles/10.1186/s12864-020-06922-2

Please revise the manuscript to rephrase the duplicated text, cite your sources, and provide details as to how the current manuscript advances on previous work. Please note that further consideration is dependent on the submission of a manuscript that addresses these concerns about the overlap in text with published work.

Response: We have rephrased the duplicated text in revised throughout the manuscript.

Reviewers' comments:

Reviewer's Responses to Questions

Comments to the Author

1. Is the manuscript technically sound, and do the data support the conclusions?

Reviewer #1: No

Reviewer #2: Partly

Reviewer #3: Yes

Response: We edited revised manuscript to solve these issues.

2. Has the statistical analysis been performed appropriately and rigorously?

Reviewer #1: No

Reviewer #2: Yes

Reviewer #3: Yes

3. Have the authors made all data underlying the findings in their manuscript fully available?

Reviewer #1: Yes

Reviewer #2: No

Reviewer #3: No

Response: Data are available online

4. Is the manuscript presented in an intelligible fashion and written in standard English?

Reviewer #1: No

Reviewer #2: No

Reviewer #3: Yes

Response: We edited and rearranged text in revised manuscript to solve these issues.

5. Review Comments to the Author

Reviewer #1: The authors proposed to analyse the genetic diversity present in a collection of 383 accessions of Brassica napus representing genotypes present worldwilde and aims at characterizing the genepool present in US breeding programs. This seems quite interesting and the size of of the populations should be sufficient to draw interesting conclusions. However hypotheses and research questions are not clearly expressed at the end of the introduction, and too many extra data are provided and not used in the manuscript (for instance flowering time data). Moreover, many analyses are performed , but may be too much without really refering to a global strategy and the global aim of each batch of analyses.

Detailed analysis and questions are reported in the attached file

A rapid check at table S1 showed some problems considering geogagraphic origins of different accessions

for instance Lembkes, Krapphauser, fertodi => Germany instead of South Korea

Dramor (Poland) should be Darmor (France) I think

Bienvenu (USA) should be Bienvenu (France)

Jet-neuf (Canada) should be Jet-neuf (France)

.... I didn't check carefully all accessions but these mistakes may lead to inappropriate interpertations of subpopulations composition

Response: Thank you for pointing these out. In revised manuscript, we have tried to incorporate the suggestions of the reviewer to make the hypotheses and research questions more clear (at later part of introduction). We remove the flowering time data and clarified how the core collection was obtained (Line: 107-116 in manuscript). We think performed analyses strengthen each other as well as the objective. 

In case of table S1, data was provided according to USDA (GRIN-Global) website (source: https://npgsweb.ars-grin.gov/gringlobal/search). We did a google search to know the origin of many cultivars and failed to get the information. USDA-GRIN recorded many genotypes on the basis of country of collection – and we used this recorded information in this manuscript. Therefore, to clear it out, we have written as “Country or origin/collected”. 

Reviewer #2: The manuscript describes an attempt to study genetic diversity and population differentiation inherent in a germplasm assemblage of Brassica napus. Study is of interest for plant breeders engaged in the improvement of this important oilseed crop. Abstract and introduction are very poorly written. The method and material section must include some information about the procedure followed to maintain germplasm lines used for genotyping. In the absence of any information, one would suspect these to be the products of open pollination. Inherent heterozygosity can be a problem for genetic studies in any often cross-pollinated crop like B.napus. Filtering out of a large proportion of heterozygous SNPs may be the cause for the retention of only 8,502 SNPs out 497336 discovered initially. Structure analysis shows lot of admixing. Heterozygosity inherent in the germ plasm will affect all genetic inferences e.g. polymorphic loci, population variation, LD, genetic differentiation and even genetic diversity. It is always better to use homozygous lines. One may not like to use the term core collection for any germplasm assemblage. Authors may like to provide reference or provide details regarding method used to extract the core ( if it is actually a core collection) from a large germplasm base. In spite of these limitations, it was interesting to find clear geographic and growth habit related clustering from STRUCTURE analysis. The manuscript also suffers from narrative coherence.

Response: Thanks for your critical inputs. In revised manuscript, we have made the changes according to reviewer’s suggestions. We rewrite the introduction section to make it clear. To improve the narrative coherence, we edited almost all section in revised manuscript. 

How core collection has been obtained and maintained, was described in materials and methods section (Line 107 to 116).

SNP number reduced, because of following strict filtering criteria (Line 134 to 139)

Reviewer #3: Manuscript PONE-D-21-09997 describes genetic diversity, population structure and LD in Brassica napus - the second major oilseed crop grown worldwide. Here are some comments:

This study does not provide details on sampling (how many plants were sampled for DNA isolation, how SNPs were scored codominant/dominant; how heterozygosity was handled etc). I could not see genotypic data of 383 accessions. Have authors submitted genotypic data to any public database repository or provided in a supplementary Table?

Response: Thank you for pointing these points. In revised manuscript, we have incorporated the information according to reviewer’s suggestion.

For DNA extraction, we used three leaf samples per genotype (Line 119)

For diversity analysis we do not need to score the codominant/dominant markers. To run structure, we scored the homozygous major, homozygous minor and heterozygous SNP as 1, 0 and 0.5 respectively using Tassel. 

Data availability: GBS and SNP data are available at: PRJNA687906 (https://www.ncbi.nlm.nih.gov/biosample/17159566) and PRJEB42419 (https://www.ebi.ac.uk/eva/?eva-study=PRJEB42419), respectively.

Reviewer #3: L172-173: Table 1 and Fig 1 does not support this claim.

Response: While we appreciate the reviewer’s feedback, we respectfully disagree. The SNP density was highest on chromosome A7 as every 71.1 kb distance contains at least one SNP and was lowest on chromosome C9 as every 134.5 kb distance contains at least one SNP. 

Reviewer #3: L191-192: It is worthwhile to provide statistics of transitions and transversions across different chromosomes

Response: As our primary objective is not to describe the SNPs in details, we think statistics of transitions and transversions across genome is enough to give an idea to the reader.

Reviewer #3: L340: GBS is one of the approaches (Illumina SNP, resequencing, sequencing and DArTseq)

Response: Yes. 

Reviewer #3: L360: There are no wild accessions of B. napus.

Response: Yes, we agree the reviewer’s comment. We deleted this information from revised manuscript

Reviewer #3: L460: Favorable alleles?

Response: We changed positive alleles to favorable alleles.

Reviewer #3: Fig 3: How much genetic variation is explained by PC1 and PC2? What these lines indicate?

Response: PC1 and PC2 explained 13.50% and 7.22% of variation respectively. Lines indicate regression. (Line 226)

Reviewer #3: Fig4: I could not see bootstrap values here. Please label all accessions (name or serial number 1 to 383 and support with supplementary Table)

Response: bootstrap value added. But did not add label as it make the figure congested. 

Reviewer #3: Figure 5 is not legible. You may move to supplementary information.

Response: We would like to keep it as it is for the better understanding of readers.

Minor comments

Reviewer #3: L52: Use either rapeseed or canola (it is already described in L49)

Response: In revised manuscript, we used rapeseed. 

Reviewer #3: L72: There are several studies to support collinearity between Arabidopsis and Brassica (see publications of Parkin and Chris Pires groups, https://doi.org/10.1111/pce.12644 etc)

Response: Yes.

Reviewer #3: L527: There are several reports. You meant NSDU accessions?

Response: Yes, we mean NDSU accessions i.e. NDSU advanced breeding lines.

---

## [Decision Letter · Decision Letter 1]

21 Oct 2021

PONE-D-21-09997R1Linkage disequilibrium and population structure in a core collection of Brassica napus (L.)PLOS ONE

Dear Dr. Rahman,

Thank you for submitting your manuscript to PLOS ONE. After careful consideration, we feel that it has merit but does not fully meet PLOS ONE’s publication criteria as it currently stands. Therefore, we invite you to submit a revised version of the manuscript that addresses the points raised during the review process.

We look forward to receiving your revised manuscript.

Kind regards,

Tzen-Yuh Chiang

Academic Editor

PLOS ONE

Reviewers' comments:

Reviewer's Responses to Questions

**Comments to the Author**

1. If the authors have adequately addressed your comments raised in a previous round of review and you feel that this manuscript is now acceptable for publication, you may indicate that here to bypass the “Comments to the Author” section, enter your conflict of interest statement in the “Confidential to Editor” section, and submit your "Accept" recommendation.

Reviewer #2: All comments have been addressed

Reviewer #4: (No Response)

2. Is the manuscript technically sound, and do the data support the conclusions?

Reviewer #2: Yes

Reviewer #4: No

3. Has the statistical analysis been performed appropriately and rigorously? 

Reviewer #2: Yes

Reviewer #4: N/A

4. Have the authors made all data underlying the findings in their manuscript fully available?

Reviewer #2: Yes

Reviewer #4: Yes

5. Is the manuscript presented in an intelligible fashion and written in standard English?

Reviewer #2: Yes

Reviewer #4: Yes

6. Review Comments to the Author

Reviewer #2: (No Response)

Reviewer #4: (No Response)

7. PLOS authors have the option to publish the peer review history of their article (what does this mean?). If published, this will include your full peer review and any attached files.

Reviewer #2: No

Reviewer #4: **Yes: **Rudolph Fredua-Agyeman

---

## [Author Response · Author response to Decision Letter 1]

29 Nov 2021

The manuscript describes studies the population structure and genetic diversity of 383 Brassica napus (canola) accessions using 8502 SNP. The authors inferred six groups based on STRUCTURE, principal component analysis and NJ analyses. The manuscript is well written. However, the STRUCTURE analysis was not comprehensive enough. In addition, the interpretation of the results was not consistent with the Figures 2 and 3. Therefore, I recommend rejection of the MS and re-consideration for publication after major revisions.

Major issues

1. Introduction

The literature cited by the authors are dated (2012 to 2014). More recent population studies on B. napus and other Brassica exist e.g., An et al. (2019) Nature Communications; Lu et al. (2019) Nature Communications and Yu et al (2021) BMC Genomics. The authors should include how their study will add to the current knowledge on genetic diversity of canola.

Response: We have added the references mentioned by the reviewer in revised manuscript. The study was done to exploit the diversity of NDSU canola core collection. The results will be used to enrich NDSU canola parental stock.

2. Materials and Methods

The authors run the STRUCTURE program using the admixture model at 10,000 burning period and 50,000 Monte Carlo Markov Chain. Yu et al (2021) showed that Puechmaille (2016) method (MedMedK, MedMeaK, MaxMedK and MaxMeaK statistics) was a better estimator of the number of clusters compared to the Evanno et al. (2005) method (DeltaK statistics) which was used by the authors. The authors should re-run STRUCTURE under different parameters to verify that the inference on population sub-populations is robust.

Response: We re-ran the structure analysis according to reviewer suggestion. We discussed it in methodology section (Line151 to line160). 

3. Results

(a) A requirement for the STRUCTURE program is that markers be unlinked. Inclusion of multiple closely linked markers may have a large effect on calculation of population structure. Also, it is not known whether these SNPs are in non-coding region or are synonymous, to eliminate the bias introduced by selection when inferring population structure. The authors should comment on the appropriateness of using the selected SNP markers for their study.

Response: To break the linkage between markers we considered the markers which are located > 1000bp from each other (L145 to L146). We didn’t consider whether SNPs are synonymous or non- synonymous as dN/dS ratio is insensitive to selection pressure in intraspecific individuals or individual from same population (Kryazhimskiy and Plotkin, 2008). 

(b) The authors did not interpret their STRUCTURE and PCA data correctly:

(i) Figure 2 shows that the maximum peak is at K=3. The second and smaller peak is at K=6. Therefore, the number of inferred clusters using STRUCTURE was likely 3 and not 6.

(ii) Figure 3-The Principal component analysis shows 3 clusters and not 6. Group 1-red (winter Europe),

Group 2 -green and blue (spring_mixed origin and spring_NDSU) and Group 3-violet and yellow (Rutabaga, winter and semi winter Asia)

NB: the miscellaneous group (black) which are scattered is in all 3 groups cannot be considered as a different subgroup (Authors own admission Lines 220-221).

Response: In revised manuscript, we corrected and discussed the figures according to reviewer suggestion (L218 to L237).

(c) The authors should indicate the NJ using rooted and unrooted trees. The authors should also indicate UPGMA rooted and unrooted trees.

Response: Unrooted NJ tree was used. We belief that Structure, PCA and NJ tree already addressed our targets. 

(d) Please could the numbers on the trees be related back to the accession identifiers?

Response: Yes, We hope the provided new picture will clarify the issue.

(e) Some accessions had the same name or variation of it but different accession number. Some accessions were also advanced breeding lines. Can the authors comment on the genetic similarity of such accessions in the results section? Will these influence the calculation of population structure?

Response: Before formulating the core collection, we discarded the accession, which has same name with different PI number or same PI number with different name. So that we did not have any accession like that. The similarity among advanced breeding lines as well as among other genotypes has been discussed in discussion section. 

Discussion

The authors need to be update after the suggested changes.

Response: We updated the discussion section.

Minor issues

1. Line 112-Did the authors use leaves from single plants or bulk plants?

Response: Bulk plants

2. Line 136-reword the core collection was divided into subgroups-Do you mean the number of sub-groups were inferred using STRUCTURE.

Response: Rewording done in revised manuscript.

Reference

Kryazhimskiy, S., and J.B. Plotkin. 2008. The population genetics of dN/dS. PLoS Genet. 4(12): e1000304.

---

## [Decision Letter · Decision Letter 2]

13 Jan 2022

PONE-D-21-09997R2Linkage disequilibrium and population structure in a core collection of Brassica napus (L.)PLOS ONE

Dear Dr. Rahman,

Thank you for submitting your manuscript to PLOS ONE. After careful consideration, we feel that it has merit but does not fully meet PLOS ONE’s publication criteria as it currently stands. Therefore, we invite you to submit a revised version of the manuscript that addresses the points raised during the review process. Please submit your revised manuscript by Feb 27 2022 11:59PM. If you will need more time than this to complete your revisions, please reply to this message or contact the journal office at plosone@plos.org. Please include the following items when submitting your revised manuscript:A rebuttal letter that responds to each point raised by the academic editor and reviewer(s). You should upload this letter as a separate file labeled 'Response to Reviewers'.A marked-up copy of your manuscript that highlights changes made to the original version. You should upload this as a separate file labeled 'Revised Manuscript with Track Changes'.An unmarked version of your revised paper without tracked changes. You should upload this as a separate file labeled 'Manuscript'.

We look forward to receiving your revised manuscript.

Kind regards,

Tzen-Yuh Chiang

Academic Editor

PLOS ONE

Reviewers' comments:

Reviewer's Responses to Questions

**Comments to the Author**

1. If the authors have adequately addressed your comments raised in a previous round of review and you feel that this manuscript is now acceptable for publication, you may indicate that here to bypass the “Comments to the Author” section, enter your conflict of interest statement in the “Confidential to Editor” section, and submit your "Accept" recommendation.

Reviewer #2: All comments have been addressed

Reviewer #5: (No Response)

2. Is the manuscript technically sound, and do the data support the conclusions?

Reviewer #2: Yes

Reviewer #5: Yes

3. Has the statistical analysis been performed appropriately and rigorously? 

Reviewer #2: Yes

Reviewer #5: No

4. Have the authors made all data underlying the findings in their manuscript fully available?

Reviewer #2: Yes

Reviewer #5: Yes

5. Is the manuscript presented in an intelligible fashion and written in standard English?

Reviewer #2: Yes

Reviewer #5: No

6. Review Comments to the Author

Reviewer #2: The authors have revised the manuscript as per the suggestions by the reviewers. The revised manuscript can be accepted for publication.

Reviewer #5: Review of Rahman et al.

Rahman et al. analyzed the genotypes of 383 germplasms of Brassica napus, a part of which consists of NDSU core collection using the genotype by sequencing method. They identified population structures and evaluated the level of linkage disequilibrium. They also observed that the C genome have much higher level of LD than the A genome.

This manuscript has been revised twice, but this is the first time for me to review the paper. I would agree that the manuscript would provide valuable information for this research area and the data presented here are mostly well analyzed, but the manuscript lacks coordination in writing and still has a lot of typos, some of which would be described later. I therefore cannot recommend the manuscript to be published in PLOS ONE in its current form.

Major comments

1. Although the introduction part is very detailed. Some important information seems missing. In particular, it is not clear how the authors defined the five subpopulations is not clear to me. There is no description for rutabaga type, which would be difficult to understand for non-specialist of B. napus.

2. I initially thought that 8502 high-quality SNPs out of 497,336 unfiltered SNPs are too small, but probably the number is the one after the thinning. Please describe the number of high-quality SNPs before the thinning.

3. There are several comparisons of statistics, but there are little statistical tests. For example, in the line 193, the authors stated that ts/tv ratio in A genome was higher than that in C genome, but the difference was not properly tested. So many of the analyses were more or less too descriptive.

4. In the STRUCTURE anslysis, I do not get what is the meaning of “accession assigned/unassigned”. STRUCTURE usually does not require any a priori assumption, in my understanding. Please add more description in the method section.

5. In the line 313, the meanings of “the mean linked LD”, “mean unlinked LD”, and “loci pair under linked LD”, are not clear to me. In the method section, the mean lined LD was calculated using SNPs with r2<0.2 (line 179), but it means they are unlinked. In addition, % is not added to 0.44, 0.02? It is somewhat confusing.

6. In the line 401, the authors mentioned that the higher transition to transversion ratio is due to natural selection, but I do not agree with it. The pattern is ubiquitously found in many organisms including humans, and in non-coding regions, and people usually think that this is due to higher mutation rate of transition than transversion.

7. The sentence in the line 427-428 seems contradicting to the previous sentence. Probably the words, “homogeneity of the diversity indices” is not appropriate.

8.In the line 471, NJ is certainly distance-based method, but PCA is not.

9.In the line 524, the authors mentioned that the level of LD in B. napus is low compared with the other crops, but there is no quantitative statement. I also wonder whether authors used the thinned data to calculate the mean LD. If ones remove the linked SNPs from the analysis, the level of LD obviously decreases.

Minor comments,

1. L143: row SNPs would be raw SNPs

2. L153: burn-in length

3. L405: Rephrase the words “moderate or low informative”.

4. L446: the word “type” is not necessary?

5. The assignment of Ks in Figure 4 is not clear. Are they determined using the presented tree?

7. PLOS authors have the option to publish the peer review history of their article (what does this mean?). If published, this will include your full peer review and any attached files.

Reviewer #2: No

Reviewer #5: No

---

## [Author Response · Author response to Decision Letter 2]

25 Jan 2022

6. Review Comments to the Author

Reviewer #2: The authors have revised the manuscript as per the suggestions by the reviewers. The revised manuscript can be accepted for publication.

Response: Thank you very much for accepting the manuscript.

Reviewer #5: Review of Rahman et al.

Rahman et al. analyzed the genotypes of 383 germplasms of Brassica napus, a part of which consists of NDSU core collection using the genotype by sequencing method. They identified population structures and evaluated the level of linkage disequilibrium. They also observed that the C genome have much higher level of LD than the A genome.

This manuscript has been revised twice, but this is the first time for me to review the paper. I would agree that the manuscript would provide valuable information for this research area and the data presented here are mostly well analyzed, but the manuscript lacks coordination in writing and still has a lot of typos, some of which would be described later. I therefore cannot recommend the manuscript to be published in PLOS ONE in its current form.

Major comments

1. Although the introduction part is very detailed. Some important information seems missing. In particular, it is not clear how the authors defined the five subpopulations is not clear to me. There is no description for rutabaga type, which would be difficult to understand for non-specialist of B. napus.

Response: We added description of rutabaga type in introduction (line 68 to 74). We assumed the subpopulation of collection according to their type and origin. Details has been added in materials and method section (line 129 to 134). 

2. I initially thought that 8502 high-quality SNPs out of 497,336 unfiltered SNPs are too small, but probably the number is the one after the thinning. Please describe the number of high-quality SNPs before the thinning.

Response: We added the number (line 151 to 164) in materials and method section

3. There are several comparisons of statistics, but there are little statistical tests. For example, in the line 193, the authors stated that ts/tv ratio in A genome was higher than that in C genome, but the difference was not properly tested. So many of the analyses were more or less too descriptive.

Response: ts/tv ratio is a simple descriptive statistics and ratio of two numbers. It does not need any statistical test. Other statistical test such as Tajima's D (line 277: 1000 permutations), AMOVA (line 291: 1023 permutations, p ˂ 0.001), Fst comparisons (line 296: 1000 permutations, p ˂ 0.01), NJ (line 251: 1000 bootstraps) has been performed accordingly. Structure analysis has been done in details. 

4. In the STRUCTURE analysis, I do not get what is the meaning of “accession assigned/unassigned”. STRUCTURE usually does not require any a priori assumption, in my understanding. Please add more description in the method section.

Response: Yes, structure does not require any a priori assumption. But as we divided the collection into subpopulation according to their type and origin, that’s why we ran structure considering accession assigned to specific subpopulation and accession unassigned to specific subpopulation to determine the exact cluster number of the collection. Details added in material and method section (line 166 to 182). 

5. In the line 313, the meanings of “the mean linked LD”, “mean unlinked LD”, and “loci pair under linked LD”, are not clear to me. In the method section, the mean lined LD was calculated using SNPs with r2<0.2 (line 179), but it means they are unlinked. In addition, % is not added to 0.44, 0.02? It is somewhat confusing.

Response: That was typo. We changed it accordingly (line 197 to 205 and line 335 to 354). 

6. In the line 401, the authors mentioned that the higher transition to transversion ratio is due to natural selection, but I do not agree with it. The pattern is ubiquitously found in many organisms including humans, and in non-coding regions, and people usually think that this is due to higher mutation rate of transition than transversion.

Response: We agreed to reviewer comment and deleted the information (line 431-432). 

7. The sentence in the line 427-428 seems contradicting to the previous sentence. Probably the words, “homogeneity of the diversity indices” is not appropriate.

Response: We agreed to reviewer comment and deleted the information (line 457 to 459).

8. In the line 471, NJ is certainly distance-based method, but PCA is not.

Response: We agreed to reviewer comment and changed accordingly (line 503 to 504).

9. In the line 524, the authors mentioned that the level of LD in B. napus is low compared with the other crops, but there is no quantitative statement. I also wonder whether authors used the thinned data to calculate the mean LD. If ones remove the linked SNPs from the analysis, the level of LD obviously decreases.

Response: We modified the information according to reviewer comments (line 558 to 561 and line 574 to 577)

Minor comments,

1. L143: row SNPs would be raw SNPs

2. L153: burn-in length

3. L405: Rephrase the words “moderate or low informative”.

4. L446: the word “type” is not necessary?

Response: We tried our best to correct typos and changed accordingly (line 151, line 161, line 411 and line 450)

5. The assignment of Ks in Figure 4 is not clear. Are they determined using the presented tree?

Response: Each branch is color-coded according to genotype belongs to subpopulation P1 to P5. Genotypes were grouped into three clusters by dividing the tree using black solid lines according to structure output (line 250 to 253).

---

## [Decision Letter · Decision Letter 3]

15 Feb 2022

Linkage disequilibrium and population structure in a core collection of Brassica napus (L.)

PONE-D-21-09997R3

Dear Dr. Rahman,

We’re pleased to inform you that your manuscript has been judged scientifically suitable for publication and will be formally accepted for publication once it meets all outstanding technical requirements.

Kind regards,

Tzen-Yuh Chiang

Academic Editor

PLOS ONE

Additional Editor Comments (optional):

Reviewers' comments:

Reviewer's Responses to Questions

**Comments to the Author**

1. If the authors have adequately addressed your comments raised in a previous round of review and you feel that this manuscript is now acceptable for publication, you may indicate that here to bypass the “Comments to the Author” section, enter your conflict of interest statement in the “Confidential to Editor” section, and submit your "Accept" recommendation.

Reviewer #5: All comments have been addressed

Reviewer #6: All comments have been addressed

2. Is the manuscript technically sound, and do the data support the conclusions?

Reviewer #5: Yes

Reviewer #6: Yes

3. Has the statistical analysis been performed appropriately and rigorously? 

Reviewer #5: Yes

Reviewer #6: Yes

4. Have the authors made all data underlying the findings in their manuscript fully available?

Reviewer #5: Yes

Reviewer #6: No

5. Is the manuscript presented in an intelligible fashion and written in standard English?

Reviewer #5: Yes

Reviewer #6: Yes

6. Review Comments to the Author

Reviewer #5: This manuscript is a revised manuscript that I have reviewed before. The authors addressed all the issues I raised so the manuscript is ready for publication.

Reviewer #6: (No Response)

7. PLOS authors have the option to publish the peer review history of their article (what does this mean?). If published, this will include your full peer review and any attached files.

Reviewer #5: No

Reviewer #6: **Yes: **Reza Talebi

---

## [Editor Report · Acceptance letter]

21 Feb 2022

PONE-D-21-09997R3 

Linkage disequilibrium and population structure in a core collection of *Brassica napus* (L.) 

Dear Dr. Rahman:

I'm pleased to inform you that your manuscript has been deemed suitable for publication in PLOS ONE. Congratulations! Your manuscript is now with our production department. 

Kind regards, 

on behalf of

Dr. Tzen-Yuh Chiang 

Academic Editor

PLOS ONE